# Mitigating the Contractivity Trap in Diffusion ODEs via Stein Stabilization

**Shigui Li** [1]   **Delu Zeng** [2]

## Abstract

A fundamental tension exists in the large-step inference of diffusion models via their deterministic probability flow ordinary differential equation (PF-ODE) trajectories, which we identify as the *contractivity trap*: efficient inference favors large step sizes, while aggressive steps and highly expressive denoisers can undermine contraction-based stability certificates for error suppression. To address this, we propose SteinDiff, a step-wise inference-time stabilization framework that employs Stein-derived corrections without requiring reference samples. Specifically, SteinDiff introduces a geometry-aware residual correction mechanism that regularizes large-step solver updates without retraining. To this end, we derive a closed-form Stein correction coefficient for step-wise solver adjustment, enabling reference-free adaptation to local data geometry. We further establish a score-controlled perturbation bound under distributional shifts and provide a complementary Stein perspective on EDM-style parameterizations. Extensive experiments demonstrate that SteinDiff mitigates severe artifacts and improves generative quality across large-step inference settings.

## 1. Introduction

Diffusion models (DMs) have emerged as a powerful approach to generative modeling, demonstrating strong performance in high-fidelity image synthesis, text-to-image generation, and other domains (Sohl-Dickstein et al., 2015; Ho et al., 2020; Song et al., 2021b; Dhariwal & Nichol, 2021; Rombach et al., 2022; Xing et al., 2024). Unlike single-pass generators such as GANs (Goodfellow et al., 2014) and VAEs (Kingma, 2013), DMs generate samples through an iterative denoising process that progressively transforms noise into structured data. This iterative paradigm provides advantages: training stability, high sample quality, and robust mode coverage (Song & Ermon, 2020; Kingma et al., 2021; Karras et al., 2022).

Despite these advantages, diffusion inference remains computationally expensive, typically requiring hundreds of function evaluations (NFE) for high-quality samples (Sohl-Dickstein et al., 2015; Ho et al., 2020). ODE-based samplers reduce this cost by following deterministic PF-ODE trajectories, leading to a series of efficient solvers and predictor-corrector schemes (Song et al., 2021b;a; Liu et al., 2022; Lu et al., 2022; Zhao et al., 2023; Zhang & Chen, 2023; Lu et al., 2025). However, aggressive few-step inference can amplify local prediction and discretization errors. From a stability perspective, contractivity of the discretized update operator $T_{\theta}$ provides a useful sufficient certificate for step-wise error suppression: $\| T_{\theta}(\boldsymbol{x}) - T_{\theta}(\boldsymbol{y}) \| \leq L \|\boldsymbol{x} - \boldsymbol{y}\|$ with $L < 1$. This condition offers a simple analytical lens for understanding whether perturbations are damped or amplified across solver updates. In the large-step regime, expressive denoisers and aggressive step sizes make such contraction-based certificates difficult to satisfy. We refer to this certificate breakdown as the *contractivity trap*, which highlights a practical regime where local perturbations may be amplified along denoising trajectories and can lead to unstable updates and degraded sample quality (Figure 1, left). To mitigate this failure mode, we propose *SteinDiff*, a principled inference-time stabilization framework based on reference-free Stein corrections (Figure 1, right). Instead of imposing rigid constraints on the model architecture or step sizes, SteinDiff introduces a geometry-aware residual correction to regularize large-step solver updates. The correction is derived from a step-wise expected squared-error criterion with respect to the latent clean target. Under the exact continuous-time forward Gaussian coupling, Stein's identity transforms the unknown clean-target term into a tractable estimator involving batch statistics and a divergence term. This yields a training-free correction mechanism that can be integrated into existing ODE solvers.

In this work, we propose a Stein-based stabilization framework for large-step ODE inference, mitigating the limitations of contraction-based certificates. Extensive experiments demonstrate the effectiveness of SteinDiff for large-step inference. Our main contributions are:

[1]School of Mathematics, South China University of Technology, Guangzhou, China; [2]School of Electronic and Information Engineering, South China University of Technology, Guangzhou, China. Correspondence to: Delu Zeng <dlzeng@scut.edu.cn>.

*Proceedings of the 43rd International Conference on Machine Learning*, Seoul, South Korea. PMLR 306, 2026. Copyright 2026 by the author(s).

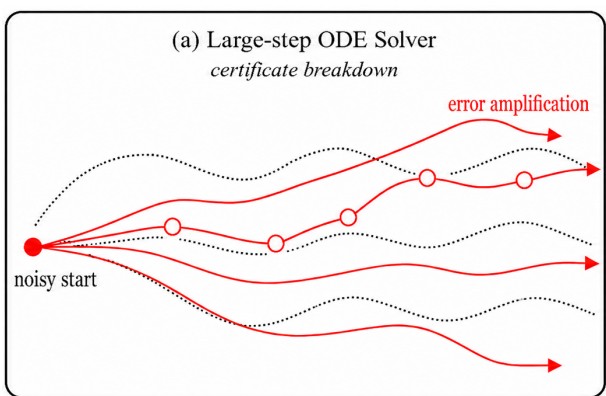 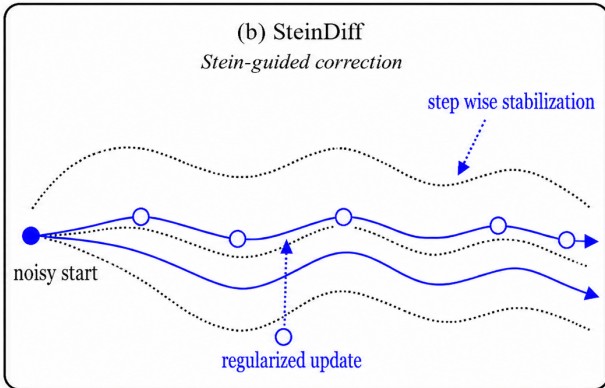

*Figure 1.* Illustration of denoising trajectories with and without principled stabilization. Efficient ODE solvers often fail to maintain contraction-based stability certificates ($L_T < 1$) due to aggressive step sizes and highly expressive denoisers, leading to compounded error accumulation and trajectory divergence (Left). SteinDiff mitigates this issue by applying a Stein-guided correction to regularize large-step solver updates and improve trajectory consistency (Right).

- We identify the *contractivity trap* as a contraction-certificate failure mode in efficient ODE solving of DMs, and propose a geometry-aware Stein correction framework for large-step solver updates.

- We derive a reference-free correction estimator via Stein's identity under the continuous-time forward Gaussian coupling, providing a principled step-wise correction toward the latent clean target.

- We analyze the perturbation stability of the Stein correction coefficient under score-controlled distributional shifts, and provide a complementary Stein perspective on EDM-style parameterizations.

## 2. Related Work

Diffusion models are a class of powerful generative models that excel at generating high-quality samples through denoising refinement, grounded in a well-established theoretical framework (Jarzynski, 1997; Neal, 2001; Sohl-Dickstein et al., 2015). Song et al. Song & Ermon (2019; 2020) developed modeling techniques using score matching, while Ho et al. (Ho et al., 2020) reformulated DMs into a practical and tractable framework. Song et al. (Song et al., 2021b) unified score-based models and DMs using stochastic differential equations (SDEs), forming the principal framework. However, DMs face the significant challenge of balancing efficiency and quality. Various methods have recently emerged to address this challenge.

Training-based methods accelerate DMs through post-training or new modeling strategies. Latent DMs (Rombach et al., 2022) boost efficiency by operating in lower-dimensional spaces. Progressive distillation (Salimans & Ho, 2022; Meng et al., 2023) and Consistency Models (CMs) (Song et al., 2023; Luo et al., 2023) achieve few-step generation through knowledge distillation and self-consistency

learning, while adversarial training is explored for the speed-quality trade-off (Xiao et al., 2022). Flow matching (FMs) (Lipman et al., 2023; Liu et al., 2023; 2024) optimizes generation paths by learning velocity fields, while EDM (Karras et al., 2022) improves samples using optimized weighting schemes. Shortcut models (Frans et al., 2025) achieve efficiency by step-aware network learning. Additional methods are also explored in (Watson et al., 2022; Wang et al., 2023; Zheng et al., 2023a; Kim et al., 2023; Zhou et al., 2024a; Kim et al., 2024; Wimbauer et al., 2024; Ma et al., 2024b; Zhou et al., 2024b; Karras et al., 2024b; Sauer et al., 2024; Karras et al., 2024a; Zhang et al., 2024; Ma et al., 2024a; Zhang et al., 2025; Tong et al., 2025; Geng et al., 2025).

Inference-focused methods improve DMs by optimizing the inference process without retraining. DDIM (Song et al., 2021a) establishes deterministic sampling using non-Markovian processes. Building on deterministic ODEs, e.g., PNDMs (Liu et al., 2022), denoising solvers have achieved significant progress: DPM-Solver (Lu et al., 2022) uses exponential integrators for acceleration, DEIS (Zhang & Chen, 2023) addresses numerical stiffness, and DPM-Solver++ (Lu et al., 2025) proposes a data-based prediction scheme. UniPC (Zhao et al., 2023) provides a predictor-corrector framework, DPM-Solver-v3 (Zheng et al., 2023b) optimizes speed with reference solution-based model statistics, while restart sampling (Xu et al., 2024b) refines generation paths using a cyclic restart mechanisms with intermediate noise injection. Recently, EVODiff (Li et al., 2025) rectifies the inference path via entropy-aware variance optimization. Additional methods include schedule optimizations (Jolicoeur-Martineau et al., 2021; Karras et al., 2022; Xue et al., 2024; Chen et al., 2024; Sabour et al., 2024), discretization techniques (Wizadwongsa & Suwajanakorn, 2023; Li et al., 2023; Gonzalez et al., 2023; Zhao et al., 2024), and parallel techniques (Shih et al., 2023; Tang et al., 2024).

**Settings and Our Contribution**   While training-based methods achieve impressive results, they suffer from expensive training costs and may sacrifice the refinement flexibility that makes DMs powerful. In contrast, inference-time methods preserve this flexibility without retraining. However, in the large-step regime, aggressive step sizes and expressive denoisers make contraction-based stability certificates difficult to satisfy. Unlike methods that rely on reference solutions, auxiliary optimization, or additional training, SteinDiff provides a reference-free principled stabilization mechanism at inference time. Specifically, it introduces a geometry-aware correction to regularize large-step solver updates. By leveraging Stein's identity under the Gaussian forward noising process, SteinDiff yields closed-form correction estimators that adapt solver updates to local data geometry. Furthermore, we analyze the robustness of the resulting empirical estimator.

## 3. Problem Setup

Diffusion models (Sohl-Dickstein et al., 2015; Ho et al., 2020; Song et al., 2021b; Karras et al., 2022) generate samples by reversing a noise-adding process. In the deterministic probability flow ODE (PF-ODE) formulation (Song et al., 2021b; Kingma et al., 2021), the generative trajectory maps an initial noise sample toward the data distribution:

$$\frac{\mathrm{d}\boldsymbol{x}}{\mathrm{d}t} = f(t)\boldsymbol{x}_t + \frac{g^2(t)}{2\sigma_t}\boldsymbol{\epsilon_\theta}(\boldsymbol{x}_t, t), \tag{1}$$

where $f(t) = \mathrm{d}\log\alpha_t/\mathrm{d}t$ and $g^2(t) = \mathrm{d}\sigma_t^2/\mathrm{d}t - 2f(t)\sigma_t^2$ are schedule-dependent functions. In practice, efficient inference often uses the data prediction parameterization

$$\boldsymbol{x_\theta}(\boldsymbol{x}_t, t) := \frac{\boldsymbol{x}_t - \sigma_t \boldsymbol{\epsilon_\theta}(\boldsymbol{x}_t, t)}{\alpha_t}, \tag{2}$$

which estimates the clean data associated with the noisy state (Kingma et al., 2021; Lu et al., 2025; Li et al., 2025).

For the theoretical derivation, we assume an ideal Gaussian coupling for the forward process:

$$\boldsymbol{x}_t = \alpha_t \boldsymbol{x}^* + \sigma_t \boldsymbol{\epsilon}, \qquad \boldsymbol{\epsilon} \sim \mathcal{N}(\boldsymbol{0}, \boldsymbol{I}), \tag{3}$$

where $\boldsymbol{x}^* \sim p_0$ denotes the latent clean data variable. This coupling is exact for the forward noising process and is used as an analytical device for deriving the correction coefficient. Practical discretized inference trajectories may deviate from this ideal coupling, which we address in the robustness analysis.

To solve Eq. (1), we discretize the PF-ODE numerically. By substituting the parameterization $\boldsymbol{x_\theta}$ into the ODE and applying the variation-of-constants formula (details in Appendix B), we define the *discretized update operator* $\mathrm{T}_{\boldsymbol{\theta}}$

mapping $\boldsymbol{x}_s$ to $\boldsymbol{x}_t$:

$$\boldsymbol{x}_t = \mathrm{T}_{\boldsymbol{\theta}}(\boldsymbol{x}_s) := \frac{\sigma_t}{\sigma_s}\boldsymbol{x}_s + \sigma_t \int_{\kappa(s)}^{\kappa(t)} \boldsymbol{x_\theta}\left(\boldsymbol{x}_{\phi(\tau)}, \phi(\tau)\right)\mathrm{d}\tau, \tag{4}$$

where $\kappa(t) := \alpha_t/\sigma_t$ is the signal-to-noise ratio parameter and $\phi(\cdot)$ denotes its inverse along the sampling trajectory.

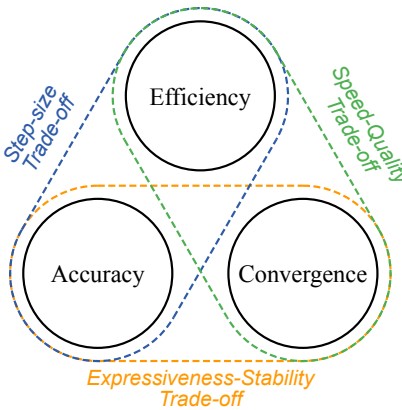

*Figure 2.* The Inference Stability Triangle.

## 4. Method

DMs generate high-quality samples by progressively mapping noise to structured data. To analyze the dynamics of large-step inference, in this section, we formalize the *Contractivity Trap*, revealing a practical tension between the high expressiveness required for diffusion models and the contraction-based stability certificate used for step-wise error suppression. To address this challenge, we propose SteinDiff, a principled framework that shifts the inference process from relying on implicit contraction-based certificates to applying explicit inference-time residual correction. Leveraging a step-wise correction structure, we derive a reference-free inference-time estimator via Stein's identity to mitigate discretization-induced deviations. Finally, we analyze the perturbation stability of the Stein correction coefficient under score-controlled distributional shift and provide a complementary Stein perspective on EDM-style parameterizations.

### 4.1. The Contractivity Trap as a Certificate Breakdown

In the few-step inference regime, contractivity of the discretized operator $\mathrm{T}_\theta$ facilitates stable error suppression. When this condition is violated, error accumulation can manifest as trajectory divergence, particularly under large step sizes. Drawing on the contraction principle as a sufficient stability criterion, we examine when $\mathrm{T}_\theta$ can be certified to satisfy

$$\|\mathrm{T}_{\boldsymbol{\theta}}(\boldsymbol{x}) - \mathrm{T}_{\boldsymbol{\theta}}(\boldsymbol{y})\| \le L_{\mathrm{T}}\|\boldsymbol{x} - \boldsymbol{y}\|, \tag{5}$$

with Lipschitz constant $L_{\mathrm{T}} < 1$. However, satisfying this certificate poses a practical tension with diffusion-generation demands. To demonstrate this, we examine the first-order discretization or DDIM, which serves as the foundation for advanced inference algorithms. Then, we have

$$
\begin{aligned}
\mathrm{F}_{\boldsymbol{\theta}}(\boldsymbol{x}_s) &= \int_{\kappa(s)}^{\kappa(t)} \boldsymbol{x}_{\boldsymbol{\theta}}\left(\boldsymbol{x}_{\phi(\tau)}, \phi(\tau)\right) \mathrm{d}\tau \\
&\approx (\kappa(t) - \kappa(s))\boldsymbol{x}_{\boldsymbol{\theta}}(\boldsymbol{x}_s, s).
\end{aligned} \tag{6}
$$

This yields the discretized inference operator: $\mathrm{T}_{\boldsymbol{\theta}}(\boldsymbol{x}_s) = \frac{\sigma_t}{\sigma_s}\boldsymbol{x}_s + \sigma_t(\kappa(t) - \kappa(s))\boldsymbol{x}_{\boldsymbol{\theta}}(\boldsymbol{x}_s, s)$. Denote $h_t := \kappa(t) - \kappa(s)$. Clearly, $h_t > 0$ for the denoising direction. By rigorous analysis detailed in Appendix C, an upper bound for the Lipschitz constant of this discretized operator is: $L_{\mathrm{T}} \leq \frac{\sigma_t}{\sigma_s} + \sigma_t h_t L_{\boldsymbol{x}_{\boldsymbol{\theta}}}$, where $L_{\boldsymbol{x}_{\boldsymbol{\theta}}}$ denotes the Lipschitz constant of the data prediction function $\boldsymbol{x}_{\boldsymbol{\theta}}(\boldsymbol{x}_t, t)$ with respect to its input $\boldsymbol{x}_t$. To certify $L_{\mathrm{T}} < 1$ using this upper bound, we require: $\frac{\sigma_t}{\sigma_s} + \sigma_t h_t L_{\boldsymbol{x}_{\boldsymbol{\theta}}} < 1$, which results in $L_{\boldsymbol{x}_{\boldsymbol{\theta}}}\left(\frac{\alpha_t}{\sigma_t} - \frac{\alpha_s}{\sigma_s}\right) < \frac{1}{\sigma_t} - \frac{1}{\sigma_s}$. However, this creates **an inherent tension triangle** as illustrated in Figure 2: **Efficiency** demands larger step sizes $h_t$ to reduce function evaluations; **Model expressiveness** requires high sensitivity (large $L_{\boldsymbol{x}_{\boldsymbol{\theta}}}$) to capture intricate distributions; and **Stable inference** requires a careful balance between these competing factors to mitigate errors.

**Proposition 4.1** (Loss of a sufficient contractivity certificate). *For the update $\mathrm{T}_{\boldsymbol{\theta}}$, the upper-bound certificate is lost when $L_{\boldsymbol{x}_{\boldsymbol{\theta}}} \geq \frac{\sigma_s - \sigma_t}{\sigma_s \alpha_t - \sigma_t \alpha_s}$. Further, since $\frac{1}{\alpha_t} > \frac{\sigma_s - \sigma_t}{\sigma_s \alpha_t - \sigma_t \alpha_s}$ for noise schedules, models with $L_{\boldsymbol{x}_{\boldsymbol{\theta}}} \geq \frac{1}{\alpha_t}$ fall outside even this lenient contractivity certificate.*

**Remark 4.2** (Vulnerability in Iterations). *The contractivity certificate can become harder to satisfy in sensitive phases of generation. Even in the late stage, as $\alpha_t \to 1$, neural network models with sufficient capacity to learn complex data distributions may exhibit large local Lipschitz constants for fine-grained discriminative capabilities (Raghu et al., 2017; Bortoli, 2022).*

*Practical Consequences.* From an operator perspective, updates that are not strictly contractive may fail to suppress local errors, and expansive updates can further amplify them. We visualize this mechanism in Figure 3 through the instructive case $\mathrm{T} = -\mathrm{I}d$, whose Lipschitz constant is 1. Although this map is non-expansive, it is not a strict contraction and can lead to indefinite oscillation rather than convergence. This instability is not merely theoretical; as empirically suggested in Figure 4, practical diffusion updates can enter locally expansive regimes. Consequently, this motivates an inference-time correction framework to mitigate local error amplification without limiting the model's capacity.

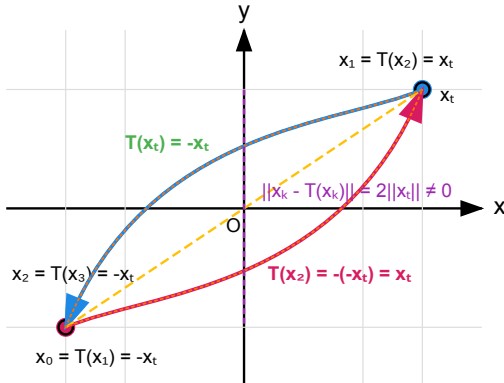

*Figure 3.* Oscillation under the non-strictly contractive map $\mathrm{T} = -\mathrm{I}d$, whose Lipschitz constant is 1. Blue and red arrows depict alternating steps ($k$ vs. $k - 1$) between $\boldsymbol{x}_t$ and $-\boldsymbol{x}_t$.

### 4.2. Towards Trajectory Stabilization

To address the contractivity trap, we rethink the inference process from the perspective of trajectory stabilization. As analyzed in Section 4.1, the high expressiveness required for DMs can push $\mathrm{T}_{\theta}$ into regimes where contraction-based error suppression is no longer certified.

To mitigate this instability, we introduce an adaptive stabilization mechanism. Rather than directly adopting the solver's candidate $\mathrm{T}_{\boldsymbol{\theta}}(\boldsymbol{x}_k)$, we formulate the updated state as a rectified estimate:

$$
\boldsymbol{x}_{k-1} = (1 - \gamma_k)\boldsymbol{x}_k + \gamma_k \mathrm{T}_{\boldsymbol{\theta}}(\boldsymbol{x}_k), \tag{7}
$$

where $\gamma_k$ serves as an adaptive correction coefficient. Unlike heuristic truncation, we seek to derive the optimal $\gamma_k$ that minimizes a step-wise estimation error relative to the latent clean target. This yields a principled inference-time correction of the solver candidate, rather than a heuristic truncation rule. By reducing the effect of aggressive large-step updates, this explicit correction framework mitigates the tension between expressiveness and stability. This point is formalized by the step-wise MSE analysis in Theorems 4.4 and 4.8, which does not require pointwise contractivity of $\mathrm{T}_{\boldsymbol{\theta}}$.

While existing parameterizations like $\boldsymbol{v}$-prediction offer improved stability over $\boldsymbol{x}$-prediction (Salimans & Ho, 2022), they do not directly provide the inference-time residual correction studied here. Thus, this explicit geometric stabilization remains crucial.

### 4.3. Step-wise Stein Correction

While the residual correction in Eq. (7) provides a structural mechanism for correcting large-step inference, the clean target $\boldsymbol{x}^*$ is not directly available during sampling. To derive a principled correction coefficient, we consider the step-wise expected squared error with respect to the latent clean

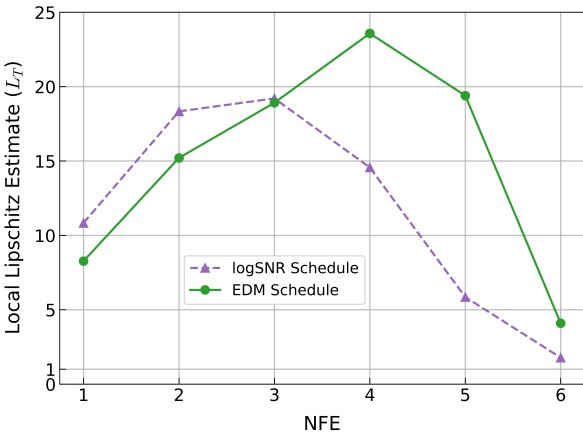
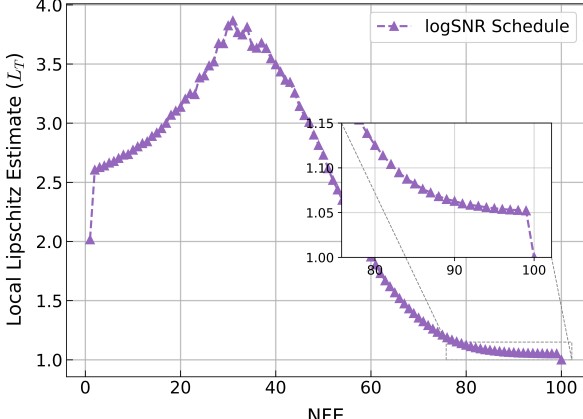

*Figure 4.* Empirical local Lipschitz estimates for efficient inference. (Left) Local expansion across schedules (NFE=6) using DPM-Solver++ for the EDM2 model. We compare local Lipschitz estimates ($L_T$) for logSNR and EDM schedules. Both schedules exhibit regions where the estimated local Lipschitz constant exceeds the strict contraction threshold ($L_T < 1$), with peaks reaching $\approx 24$. This supports the practical relevance of the contractivity trap in large-step inference. (Right) Persistence despite finer discretization (NFE=100). Even with smaller steps, the estimated operator remains near or above the strict contraction threshold for a large portion of the trajectory (see inset). This suggests that step refinement alone may not fully remove local expansion effects in practical samplers.

variable under the forward noising process:

$$\boldsymbol{x}_k = \alpha_k \boldsymbol{x}^* + \sigma_k \boldsymbol{\epsilon}, \qquad \boldsymbol{\epsilon} \sim \mathcal{N}(\boldsymbol{0}, \mathbf{I}). \qquad (8)$$

Here, $\boldsymbol{x}^* \sim p_0$ denotes the clean data variable in the forward process. This views $\boldsymbol{x}_k$ through the forward conditional Gaussian $q(\boldsymbol{x}_k|\boldsymbol{x}^*)$, which motivates our application of Stein's identity to transform the unknown clean-target term into a tractable divergence term, yielding a reference-free estimator that captures local geometric information through the divergence of the solver residual.

**Theorem 4.3** (Step-wise MSE-optimal correction). *Consider the local step-wise objective of minimizing the expected squared distance to the latent clean target:*

$$J(\gamma_k) = \mathbb{E}\big[\|(1-\gamma_k)\boldsymbol{x}_k + \gamma_k \, \mathrm{T}_{\boldsymbol{\theta}}(\boldsymbol{x}_k) - \boldsymbol{x}^*\|^2\big],$$

*where the expectation is taken under the ideal forward coupling $\boldsymbol{x}^* \sim p_0$ and $\boldsymbol{x}_k|\boldsymbol{x}^* \sim \mathcal{N}(\alpha_k \boldsymbol{x}^*, \sigma_k^2 \mathbf{I})$. Let $\boldsymbol{u}_k := \boldsymbol{x}_k - \mathrm{T}_{\boldsymbol{\theta}}(\boldsymbol{x}_k)$, $c_k := \mathbb{E}\|\boldsymbol{u}_k\|^2 > 0$. Then the minimizer of $J(\gamma_k)$ is*

$$\gamma_k^* = \frac{\mathbb{E}\langle \boldsymbol{u}_k, \, \boldsymbol{x}_k - \boldsymbol{x}^* \rangle}{\mathbb{E}\|\boldsymbol{u}_k\|^2}. \qquad (9)$$

*The complete proof is provided in Appendix D.1.*

**Theorem 4.4** (Step-wise MSE Improvement). *The vanilla ODE solver corresponds to $\gamma = 1$. The coefficient $\gamma_k^*$ from Theorem 4.3 minimizes the quadratic objective $J(\gamma)$ over $\gamma$, yielding a step-wise expected error no greater than that of the vanilla update:*

$$\mathbb{E}\big[\|\boldsymbol{x}_{k-1}^{Stein} - \boldsymbol{x}^*\|^2\big] \leq \mathbb{E}\big[\|\,\mathrm{T}_{\boldsymbol{\theta}}(\boldsymbol{x}_k) - \boldsymbol{x}^*\|^2\big], \qquad (10)$$

*with equality iff $b_k = c_k$, i.e., when the vanilla coefficient $\gamma = 1$ already minimizes the step-wise objective. The step-wise improvement is $\Delta J := J(1) - J(\gamma_k^*) = (b_k - c_k)^2/c_k \geq 0$ where $b_k := \mathbb{E}[\langle \boldsymbol{u}_k, \boldsymbol{x}_k - \boldsymbol{x}^* \rangle]$ and $c_k := \mathbb{E}[\|\boldsymbol{u}_k\|^2]$. The complete proof is provided in Appendix D.2.*

**Remark 4.5** (Clean-target alignment). *The expression of $\gamma_k^*$ shows that the correction is governed by a normalized alignment between the solver residual $\boldsymbol{u}_k = \boldsymbol{x}_k - \mathrm{T}_{\boldsymbol{\theta}}(\boldsymbol{x}_k)$ and the clean-target direction $\boldsymbol{x}_k - \boldsymbol{x}^*$. The numerator measures the component of the residual that is consistent with the desired denoising direction, while the denominator normalizes this quantity by the residual energy. Consequently, SteinDiff calibrates each inference step according to the expected alignment between the solver residual and the clean-target direction under the forward coupling.*

While Theorem 4.3 provides the theoretical optimum, Eq. (9) remains intractable during inference due to the unknown clean data $\boldsymbol{x}^*$. To derive a tractable, reference-free estimator, we use the ideal forward noising coupling $p_{0k}(\boldsymbol{x}_k|\boldsymbol{x}^*) = \mathcal{N}(\boldsymbol{x}_k; \alpha_k \boldsymbol{x}^*, \sigma_k^2 \mathbf{I})$, which is exact for the forward process (Anderson, 1982; Song et al., 2021b). Although discretized inference inevitably introduces approximation errors, this exact coupling provides a principled pathway to invoke Stein's Identity for reference-free estimation, with the resulting deviations explicitly analyzed in our subsequent robustness analysis.

**Lemma 4.6** (Stein's Identity). *Let $\boldsymbol{x} \sim \mathcal{N}(\boldsymbol{\mu}, \sigma^2 \mathbf{I})$. For any differentiable vector field $\boldsymbol{v}$ with suitable integrability: $\mathbb{E}\big[\langle \boldsymbol{v}(\boldsymbol{x}), \boldsymbol{x} - \boldsymbol{\mu} \rangle\big] = \sigma^2 \mathbb{E}\big[\nabla \cdot \boldsymbol{v}(\boldsymbol{x})\big]$. We provided the proof in Appendix D.3.*

Applying Lemma 4.6 transforms the intractable term $\mathbb{E}[\langle \boldsymbol{u}_k, \boldsymbol{x}^* \rangle]$ into a computable divergence:

**Theorem 4.7** (Reference-free estimator via Stein's identity). *Under the ideal forward coupling above, the step-wise MSE-optimal coefficient $\gamma_k^*$ admits the reference-free expression:*

$$\boxed{\gamma_k^* = \frac{\left(1 - \frac{1}{\alpha_k}\right)\mathbb{E}\langle \boldsymbol{u}_k, \boldsymbol{x}_k \rangle + \frac{\sigma_k^2}{\alpha_k}\mathbb{E}[\nabla \cdot \boldsymbol{u}_k]}{\mathbb{E}\|\boldsymbol{u}_k\|^2}}, \quad (11)$$

*where $\nabla \cdot \boldsymbol{u}_k = \sum_{i=1}^{d} \partial_{\boldsymbol{x}_k^{(i)}}(\boldsymbol{u}_k)_i$ is the divergence. The proof is given in Appendix D.4.*

**Theorem 4.8** (Step-wise error decay). *Let $\boldsymbol{u}_k = \boldsymbol{x}_k - \mathrm{T}_{\boldsymbol{\theta}}(\boldsymbol{x}_k)$, $b_k = \mathbb{E}[\langle \boldsymbol{u}_k, \boldsymbol{x}_k - \boldsymbol{x}^* \rangle]$, $c_k = \mathbb{E}[\|\boldsymbol{u}_k\|^2]$. Assume $c_k > 0$, and let $\gamma_k^* = \frac{b_k}{c_k}$. Then the exact SteinDiff update $\boldsymbol{x}_{k-1}^{\mathrm{Stein}} = \boldsymbol{x}_k - \gamma_k^* \boldsymbol{u}_k$ satisfies*

$$\mathbb{E}\left[\|\boldsymbol{x}_{k-1}^{\mathrm{Stein}} - \boldsymbol{x}^*\|^2\right] = \mathbb{E}\left[\|\boldsymbol{x}_k - \boldsymbol{x}^*\|^2\right] - \frac{b_k^2}{c_k}. \quad (12)$$

*Consequently, if $E_k = \mathbb{E}\left[\|\boldsymbol{x}_k - \boldsymbol{x}^*\|^2\right]$, $\rho_k = \frac{b_k^2}{c_k E_k}$ with $E_k > 0$, then*

$$0 \leq \rho_k \leq 1 \text{ and } E_{k-1}^{\mathrm{Stein}} = (1 - \rho_k)E_k. \quad (13)$$

*Furthermore, along the trajectory over $N$ steps,*

$$E_0^{\mathrm{Stein}} = E_N \prod_{k=1}^{N}(1 - \rho_k). \quad (14)$$

*Denote $\eta := \min\{\rho_k\}$. Then $\rho_k \geq \eta$, and we have*

$$E_0^{\mathrm{Stein}} \leq (1 - \eta)^N E_N. \quad (15)$$

*The proof is provided in Appendix D.5.*

**Corollary 4.9** (Vanilla consistency). *Under the assumptions of Theorem 4.8, the deviation between the exact SteinDiff update and the vanilla solver candidate satisfies*

$$\mathbb{E}\left[\|\boldsymbol{x}_{k-1}^{\mathrm{Stein}} - \mathrm{T}_{\boldsymbol{\theta}}(\boldsymbol{x}_k)\|^2\right] \leq \mathbb{E}\left[\|\mathrm{T}_{\boldsymbol{\theta}}(\boldsymbol{x}_k) - \boldsymbol{x}^*\|^2\right]. \quad (16)$$

*Therefore, whenever the vanilla solver candidate becomes accurate in the expected MSE sense, the SteinDiff update becomes asymptotically equivalent to that candidate. The proof is provided in Appendix D.6.*

### 4.4. A Step-wise SteinDiff Estimator

Motivated by the step-wise MSE-optimal coefficient, we present its practical inference-time estimator. $\gamma_k^*$ denotes the ideal-coupling coefficient, $\tilde{\gamma}_k$ the coefficient under the discretized sampler, and $\hat{\gamma}_k$ the practical Hutchinson-based estimate (Algorithm 1). SteinDiff does not require training

---

**Algorithm 1** A SteinDiff Correction

**Require:** State $\boldsymbol{x}_{t_k}$; Model $\boldsymbol{x}_{\boldsymbol{\theta}}$; Update operator $\mathrm{T}_{\boldsymbol{\theta}}$;
  Schedule $\{t_k, \alpha_{t_k}, \sigma_{t_k}\}$; Batch size $B$.
1: $\boldsymbol{u}_k \leftarrow \boldsymbol{x}_{t_k} - \mathrm{T}_{\boldsymbol{\theta}}(\boldsymbol{x}_{t_k})$
2: $\hat{s}_{xu} \leftarrow \frac{1}{B}\sum\langle\boldsymbol{u}_k, \boldsymbol{x}_{t_k}\rangle$; $\quad \hat{s}_{uu} \leftarrow \frac{1}{B}\sum\|\boldsymbol{u}_k\|^2$
3: $\hat{s}_{div} \leftarrow \frac{1}{B}\sum \boldsymbol{v}^\top \nabla_{\boldsymbol{x}}\boldsymbol{u}_k\boldsymbol{v}$, with $\boldsymbol{v} \sim \mathcal{N}(0, \boldsymbol{I})$
4: $\hat{\gamma}_k \leftarrow \max\left(\frac{(1-1/\alpha_{t_k})\hat{s}_{xu}+(\sigma_{t_k}^2/\alpha_{t_k})\hat{s}_{div}}{\hat{s}_{uu}}, \gamma_{\min}\right)$
5: $\boldsymbol{x}_{t_{k-1}} \leftarrow (1-\hat{\gamma}_k)\boldsymbol{x}_{t_k} + \hat{\gamma}_k \mathrm{T}_{\boldsymbol{\theta}}(\boldsymbol{x}_{t_k})$
6: **Return** $\boldsymbol{x}_{t_{k-1}}$

---

samples or reference clean data. The correction is computed from the current solver residual, with the Hutchinson estimator approximating only the divergence term. A generation batch can reduce Monte Carlo variance, but is not conceptually required by the correction mechanism.

For a generation batch of size $B$, the expectations in Eq. (11) are approximated by the empirical averages

$$\hat{s}_{xu} = \frac{1}{B}\sum_{i=1}^{B}\langle\boldsymbol{u}_k^{(i)}, \boldsymbol{x}_k^{(i)}\rangle, \; \hat{s}_{uu} = \frac{1}{B}\sum_{i=1}^{B}\|\boldsymbol{u}_k^{(i)}\|^2. \quad (17)$$

We estimate the divergence via Hutchinson's trace estimator:

$$\hat{s}_{div} = \frac{1}{B}\sum_{i=1}^{B}\boldsymbol{v}^{(i)\top}\nabla_{\boldsymbol{x}}\boldsymbol{u}_k^{(i)}\boldsymbol{v}^{(i)}, \; \boldsymbol{v}^{(i)} \sim \mathcal{N}(\boldsymbol{0}, \boldsymbol{I}). \quad (18)$$

Algorithm 1 details the implementation.

While $\gamma_k^*$ inherently assumes the exact forward Gaussian coupling, discretized inference inevitably introduces distributional shifts. We characterize the impact of this shift through a conditional perturbation bound.

**Theorem 4.10** (Perturbation of the SteinDiff estimator). *Let $p_k$ denote the marginal induced by the ideal forward coupling used in the derivation of Eq. (11), and let $\tilde{p}_k$ denote the distribution induced by the discretized sampler. Let $\gamma_k^*$ and $\tilde{\gamma}_k$ be the corresponding correction coefficients computed under $p_k$ and $\tilde{p}_k$, respectively. Define the score deviation*

$$\mathcal{S}(\tilde{p}_k, p_k) := \left(\mathbb{E}_{\tilde{p}_k}\left[\|\nabla\log\tilde{p}_k - \nabla\log p_k\|^2\right]\right)^{1/2}. \quad (19)$$

*Assume that the denominators of the two coefficients are bounded away from zero, and that $\boldsymbol{u}_k$, $\nabla \cdot \boldsymbol{u}_k$, and the functions appearing in Eq. (11) are sufficiently regular so that their expectation shifts are controlled by $\mathcal{S}(\tilde{p}_k, p_k)$. Then there exists a finite constant $C_k$, depending on these regularity constants, such that*

$$|\tilde{\gamma}_k - \gamma_k^*| \leq C_k \mathcal{S}(\tilde{p}_k, p_k). \quad (20)$$

*For EDM-style parameterization ($\alpha_k \equiv 1$), the drift-related term in the numerator of Eq. (11) vanishes, leaving only the divergence-based term. The proof is provided in Appendix D.7.*

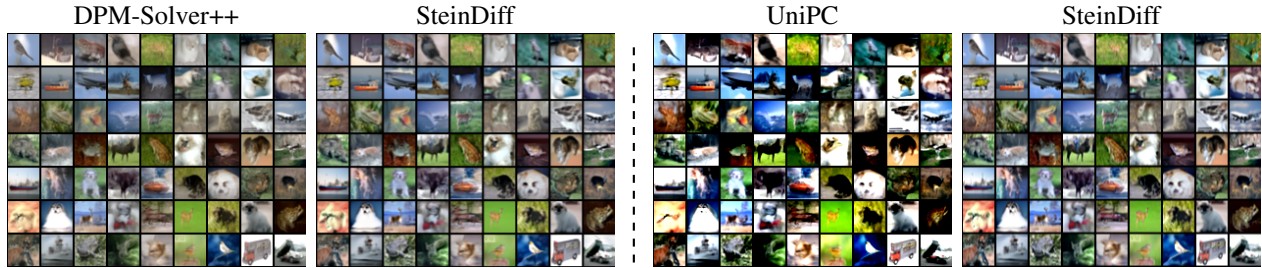

*Figure 5.* SteinDiff addresses the contractivity trap in few-step inference: at just *3 solver steps* (5 NFE), it improves few-step sampling with DPM-Solver++ and UniPC by mitigating severe artifacts and generating higher-quality samples on CIFAR-10 with the EDM model.

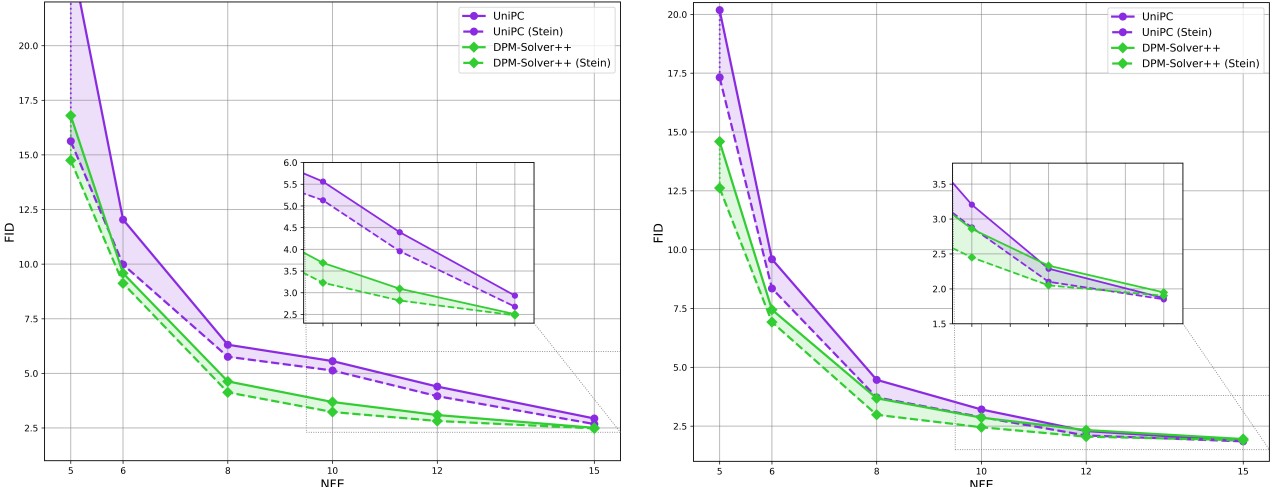

*Figure 6.* FID ↓ scores for DPM-Solver++ and UniPC using third-order solvers on ImageNet 64×64 under EDM (left) and logSNR (right) noise schedules. SteinDiff (dashed) consistently improves FID across various NFEs.

Theorem 4.10 provides a conditional perturbation guarantee showing that the correction coefficient under the discretized-sampler distribution closely approximates its ideal-coupling counterpart, provided that the induced distribution exhibits a sufficiently small score deviation from the ideal marginal. In the EDM framework, although the drift-related component in Eq. (11) vanishes, the remaining divergence-based term can still dominate the perturbation bound.

**Theorem 4.11** (Estimator perturbation around the MSE-optimal coefficient). *Let*

$$J_k(\gamma) = \mathbb{E}_{p_k}\left[\|(1-\gamma)\boldsymbol{x}_k + \gamma \,\mathrm{T}_{\boldsymbol{\theta}}(\boldsymbol{x}_k) - \boldsymbol{x}^*\|^2\right], \quad (21)$$

*and let $\gamma_k^*$ be the MSE-optimal coefficient from Theorem 4.3. For any coefficient $\bar{\gamma}_k$, we have*

$$J_k(\bar{\gamma}_k) = J_k(\gamma_k^*) + c_k(\bar{\gamma}_k - \gamma_k^*)^2, \quad c_k = \mathbb{E}_{p_k}\|\boldsymbol{u}_k\|^2. \quad (22)$$

*Consequently, the correction with $\bar{\gamma}_k$ improves over the vanilla update whenever*

$$c_k(\bar{\gamma}_k - \gamma_k^*)^2 \le J_k(1) - J_k(\gamma_k^*). \quad (23)$$

*The proof is provided in Appendix D.8.*

**Corollary 4.12** (Controlled degradation under distribution shift). *Under the assumptions of Theorem 4.10, the excess error caused by using $\tilde{\gamma}_k$ instead of $\gamma_k^*$ is controlled by*

$$J_k(\tilde{\gamma}_k) - J_k(\gamma_k^*) \le c_k C_k^2 \mathcal{S}(\tilde{p}_k, p_k)^2. \quad (24)$$

*Thus, when the score deviation is small relative to the optimality gap, the distribution-shifted correction preserves the step-wise improvement. Additional finite-batch, Hutchinson, and clipping errors can be incorporated into the total perturbation $|\hat{\gamma}_k - \gamma_k^*|$. The proof is provided in Appendix D.9.*

**Remark 4.13** (Addressing the contractivity trap). *The exact coefficient $\gamma_k^*$ provides step-wise expected MSE improvement without requiring the solver update $\mathrm{T}_{\boldsymbol{\theta}}$ to be pointwise contractive. The practical estimator preserves this improvement when the total perturbation $|\hat{\gamma}_k - \gamma_k^*|$ satisfies the gap condition in Theorem 4.11. Thus, SteinDiff addresses the contraction-certificate failure mode by optimizing a step-wise error objective rather than enforcing a Lipschitz constraint on the denoiser. It does not require extra solver NFEs, but it introduces a VJP-based divergence-estimation cost.*

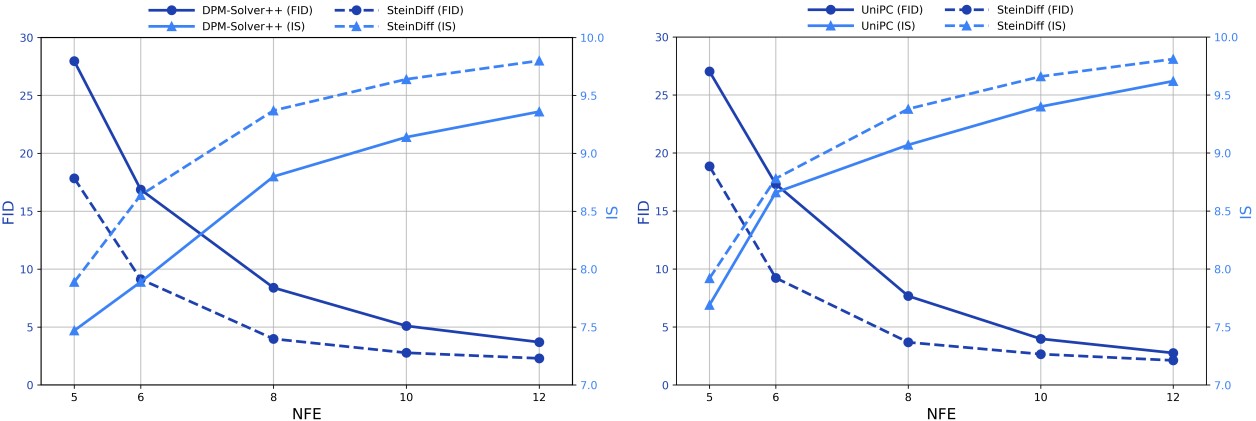

*Figure 7.* FID ↓ and IS ↑ scores vs. NFE for DPM-Solver++ (left) and UniPC (right) with/without SteinDiff on CIFAR-10 (EDM).

### 4.5. A Stein Perspective on EDM Parameterizations

The Stein framework also provides a useful lens for interpreting existing diffusion parameterizations. In particular, the step-wise optimal correction coefficient $\gamma_k^*$ reveals how different parameterizations distribute the correction burden between global signal scaling and local residual geometry. This perspective is especially relevant to EDM-style formulations (Karras et al., 2022; 2024b;a), which have been empirically observed to be robust under carefully designed noise schedules and preconditioning choices.

Specifically, the numerator of $\gamma_k^*$ decomposes into two distinct components: a drift-related term, $(1 - \frac{1}{\alpha_k})\mathbb{E}\langle \boldsymbol{u}_k, \boldsymbol{x}_k \rangle$, which reflects global signal-scaling effects, and a divergence-based geometric term, $\frac{\sigma_k^2}{\alpha_k}\mathbb{E}[\nabla \cdot \boldsymbol{u}_k]$, which captures the local residual geometry. In EDM-style parameterizations, the signal scaling is normalized by taking $\alpha_k \equiv 1$. Consequently, the signal-scaling component vanishes, and the correction coefficient reduces to the purely geometric form $\frac{\sigma_k^2 \mathbb{E}[\nabla \cdot \boldsymbol{u}_k]}{\mathbb{E}\|\boldsymbol{u}_k\|^2}$.

This simplification suggests that EDM-style parameterizations decouple step-wise trajectory correction from global signal-scaling dynamics, allowing the correction to depend primarily on the local residual geometry. From this perspective, the empirical robustness of EDM-style samplers is consistent with a geometric preference: large-step inference benefits when the update direction is governed less by global rescaling effects and more by local properties of the residual vector field.

Importantly, this observation should not be interpreted as an unconditional stability guarantee. The remaining divergence term, the quality of the learned denoiser, and stochastic errors from practical trace estimation still affect the behavior of the sampler. Rather, the Stein perspective provides an explanatory insight: EDM-style normalization suppresses a drift-related component in the optimal correction coefficient, while SteinDiff explicitly estimates and applies the remaining geometry-aware correction. This alignment suggests that, especially in aggressive few-step regimes, stabilizing local residual geometry can be more beneficial than relying solely on finer discretization or schedule refinement.

## 5. Experiments

In our experiments, we use the well-established Frechet Inception Distance (FID) and Inception Score (IS) (Heusel et al., 2017; Salimans et al., 2016) metric to measure the quality of the generated images. Furthermore, as the FID score often unfairly favors the models trained with GAN losses and penalizes the diffusion models, we consider an additional metrics of FD-DINOv2 (Stein et al., 2023), which replaces the InceptionV3 (Szegedy et al., 2016) encoder of FID by DINOv2 (Oquab et al., 2024) to better align with human perception. Furthermore, we evaluate the efficiency of our regularized inference scheme using metrics such as the number of solver calls ("Steps") and the Number of Function Evaluations (NFE); lower number metrics indicate higher inference efficiency.

*Variants.* In practice, SteinDiff can optionally incorporate a self-consistency correction mechanism to further mitigate discretization errors by leveraging look-ahead trajectory information. Specific details of this variant are provided in Appendix D.11.

*Results.* Our empirical results support the practical relevance of the *contractivity trap* perspective and show that SteinDiff consistently improves efficient ODE sampling. On CIFAR-10, SteinDiff mitigates severe artifacts at 5 NFE (Figure 5) and improves image quality across different step settings, demonstrating its effectiveness even under aggressive inference budgets. As shown in Figure 7, our method achieves better FID and IS metrics than the baseline, indi-

| 5 NFE | 20 NFE |
|:---:|:---:|

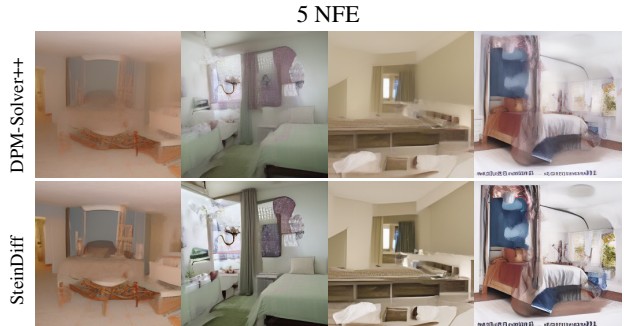 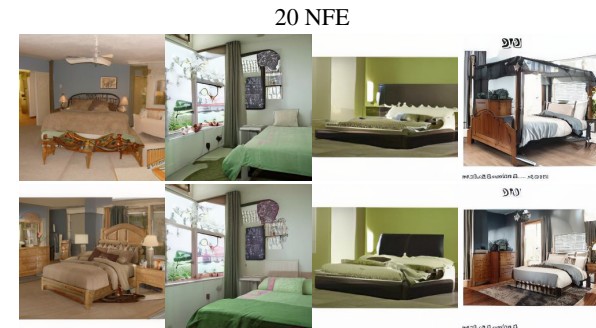

*Figure 8.* Visual comparison on 256×256 LSUN-Bedrooms: DPM-Solver++ (top) falls into the contractivity trap, while SteinDiff (bottom) overcomes it, leveraging the underlying geometric structure for efficient inference and improved quality across varying large-steps.

*Table 1.* Performance comparison on 256×256 LSUN-Bedrooms using a Latent Diffusion model. We evaluate FID scores against DPM-Solver++ across various NFEs, where lower scores indicate better image quality. The results demonstrate that our step-wise stabilization method (SteinDiff) significantly outperforms the baseline variants at all tested NFEs.

| Method | Model | NFE | | | | | | |
|---|---|---|---|---|---|---|---|---|
| | | 5 | 6 | 8 | 10 | 12 | 15 | 20 |
| DPM-Solver++ (2m) | Latent Diffusion, LSUN_beds-256 | 21.29 | 10.97 | 5.13 | 3.88 | 3.52 | 3.34 | 3.25 |
| DPM-Solver++ (3m) | | 18.61 | 8.52 | 4.15 | 3.61 | 3.43 | 3.28 | 3.17 |
| SteinDiff (SC) | | **7.64** | **4.71** | **3.72** | **3.38** | **3.01** | **2.86** | **2.77** |

cating improved robustness to step-size variations.

This trend is further observed on ImageNet 64×64 dataset, where SteinDiff provides substantial gains across different solvers, including DPM-Solver++, UniPC, and Heun. Notably, these improvements are observed under both EDM and logSNR schedules, with FID reductions of up to **45.8%** (Table 3). This consistency suggests that SteinDiff is not tied to a specific solver or noise schedule.

We further evaluate the compatibility of SteinDiff with latent-space diffusion models. On the LSUN Bedrooms 256×256 dataset, SteinDiff achieves competitive performance under the evaluated setting (Table 1). As visualized in Figure 8, the baseline DPM-Solver++ exhibits severe structural artifacts at 5 NFE, whereas SteinDiff preserves more coherent geometric structures. This qualitative result is consistent with the local expansion behavior illustrated in Figure 4. Overall, these results suggest that SteinDiff provides an effective training-free stabilization mechanism for efficient generative inference across different datasets, solvers, and model families. Additional experimental details are provided in Appendix E.

## Conclusion

In this work, we identify and formalize the *contractivity trap*—a fundamental stability problem in the efficient ODE solving of diffusion models. To overcome this limitation at inference time, we introduce *SteinDiff*, a training-free, plug-

and-play stabilization framework. By leveraging Stein's identity to convert the clean-target alignment term into a reference-free divergence correction, SteinDiff provides a reference-free geometric correction that mitigates severe structural collapse under aggressive step sizes. Crucially, this mechanism avoids additional solver NFEs and model retraining, with the extra cost mainly arising from parallelizable VJP-based divergence estimation. Furthermore, our Stein-based perspective theoretically rationalizes the empirical success of EDM parameterizations, offering a principled theoretical lens to guide the co-design of future generative architectures and their efficient inference strategies.

## Limitations and Future Work

While SteinDiff provides step-wise stabilization for efficient ODE solving, its performance remains bounded by the capacity of the pre-trained model and the variance introduced by Hutchinson-based divergence estimation. Its Monte Carlo error may slightly affect the correction coefficient, especially in small-batch or high-dimensional settings. Furthermore, the contractivity trap may become more pronounced in high-dimensional continuous spaces, where local geometric deviations can accumulate more severely under aggressive few-step inference. An important future direction is to extend this training-free stabilization framework to large-scale video generation models and investigate whether Stein-guided correction can mitigate high-frequency geometric drift during few-step inference.

## Acknowledgements

This work was supported in part by grants from National Natural Science Foundation of China (52539005), the China Scholarship Council (202306150167), the fundamental research program of Guangdong, China (2023A1515011281), Guangdong Basic and Applied Basic Research Foundation (24202107190000687), Foshan Science and Technology Research Project (2220001018608).

## Impact Statement

By enabling efficient, training-free inference, SteinDiff democratizes access to high-fidelity generative models and reduces their carbon footprint. However, because our geometric correction does not alter the fundamental data distribution, it cannot rectify inherent training data biases or mitigate misuse risks like deepfakes.

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

# A. Notation and Background for Diffusion Models

Denoising diffusion probabilistic models (DDPMs) define a forward process that adds Gaussian noise to data according to a predefined variance schedule $\{\beta_t\}_{t=1}^N$. Although originally formulated in the discrete-time setting, this transition process admits a natural continuous-time generalization (Kingma et al., 2021). In both cases, the conditional distribution of a noisy sample $\boldsymbol{x}_t$ given the clean sample $\boldsymbol{x}^*$ takes the form

$$q(\boldsymbol{x}_t|\boldsymbol{x}^*) := \mathcal{N}(\boldsymbol{x}_t; \alpha_t \boldsymbol{x}^*, \sigma_t^2 \mathbf{I}), \tag{25}$$

where $\alpha_t$ and $\sigma_t$ are deterministic functions of time controlling the signal and noise scales, respectively. In the discrete DDPM formulation, they are related to the variance schedule by $\alpha_t^2 = \prod_{s=1}^t (1-\beta_s)$ and $\sigma_t^2 = 1 - \alpha_t^2$. For discretized inference, we denote $\alpha_k := \alpha_{t_k}$ and $\sigma_k := \sigma_{t_k}$ for a chosen timestep sequence $t_k$.

To reverse the forward noising dynamics, a neural network $\boldsymbol{\epsilon\theta}(\boldsymbol{x}_t, t)$ is trained to predict the noise $\boldsymbol{\epsilon}$ added at each timestep. The training objective is simplified to minimizing the mean squared error (MSE):

$$\mathcal{L} = \mathbb{E}_{t,\boldsymbol{x}^*,\boldsymbol{\epsilon}} \left\| \boldsymbol{\epsilon} - \boldsymbol{\epsilon}_\theta(\alpha_t \boldsymbol{x}^* + \sigma_t \boldsymbol{\epsilon}, t) \right\|^2, \tag{26}$$

where $t$ is uniformly sampled from $\{1, \ldots, N\}$, $\boldsymbol{x}^* \sim q(\boldsymbol{x}^*)$ represents a sample from the data distribution, and $\boldsymbol{\epsilon} \sim \mathcal{N}(\boldsymbol{0}, \mathbf{I})$ is standard Gaussian noise. Samples are generated by iteratively applying the learned reverse transition $p_\theta(\boldsymbol{x}_{t-1}|\boldsymbol{x}_t)$.

This discrete formulation can be unified with continuous-time diffusion models via stochastic differential equations (SDEs) (Song et al., 2021b). Formally, a unified forward process for different schedules (including VP, VE, and sub-VP schedules) is formulated as:

$$d\boldsymbol{x}_t = f(t)\boldsymbol{x}_t \, dt + g(t) d\boldsymbol{w}, \quad \boldsymbol{x}^* \sim q_0(\boldsymbol{x}^*), \tag{27}$$

where the drift and diffusion coefficients are linked to the noise schedule through $f(t) := \frac{d \log \alpha_t}{dt}$ and $g^2(t) := \frac{d\sigma_t^2}{dt} - 2\frac{d \log \alpha_t}{dt}\sigma_t^2$ (Kingma et al., 2021). The corresponding reverse-time SDEs of these schedules share a unified form:

$$d\boldsymbol{x} = [f(t)\boldsymbol{x} - g(t)^2 \nabla_{\boldsymbol{x}} \log p_t(\boldsymbol{x})]dt + g(t)d\bar{\boldsymbol{w}}, \tag{28}$$

where $\bar{\boldsymbol{w}}$ is a standard Wiener process in the reverse-time direction. Its deterministic counterpart is the probability flow (PF) ordinary differential equation (ODE), which shares the same marginal distributions as the SDE. It is given by:

$$d\boldsymbol{x} = \left( f(t)\boldsymbol{x} - \frac{1}{2}g(t)^2 \nabla_{\boldsymbol{x}} \log p_t(\boldsymbol{x}) \right) dt. \tag{29}$$

Under Gaussian modeling assumptions, the score function $\nabla_{\boldsymbol{x}} \log p_t(\boldsymbol{x})$ and the noise predictor $\boldsymbol{\epsilon\theta}$ are linked by the relation $\boldsymbol{\epsilon\theta}(\boldsymbol{x}_t, t) = -\sigma_t \nabla_{\boldsymbol{x}} \log p_t(\boldsymbol{x}_t)$ (Song et al., 2021b; Karras et al., 2022). Substituting this relation into Eq. (29) leads to the diffusion ODE form used in our main text (Preliminaries 3):

$$\frac{d\boldsymbol{x}}{dt} = f(t)\boldsymbol{x}_t + \frac{g^2(t)}{2\sigma_t}\boldsymbol{\epsilon\theta}(\boldsymbol{x}_t, t). \tag{30}$$

The denoising process can be parameterized by predicting the clean data $\boldsymbol{x}^*$ from a noisy state $\boldsymbol{x}_t$ with efficient sampling advantages (Lu et al., 2025; Li et al., 2025):

$$\boldsymbol{x\theta}(\boldsymbol{x}_t, t) = \frac{\boldsymbol{x}_t - \sigma_t \boldsymbol{\epsilon\theta}(\boldsymbol{x}_t, t)}{\alpha_t}. \tag{31}$$

Substituting this parameterization into the diffusion ODE shown in Eq. (1), one has

$$\frac{d\boldsymbol{x}}{dt} = \left( f(t) + \frac{g^2(t)}{2\sigma_t^2} \right) \boldsymbol{x}_t - \alpha_t \frac{g^2(t)}{2\sigma_t^2} \boldsymbol{x\theta}(\boldsymbol{x}_t, t). \tag{32}$$

In practice, one can tailor the inference algorithms for DMs by numerically solving this diffusion ODE using discretization techniques. For any time interval $[t, s]$, the details are provided in Appendix B, we can express the discretized mapping of this ODE from $\boldsymbol{x}_s$ to $\boldsymbol{x}_t$ as the operator $\mathrm{T}_{\boldsymbol{\theta}}(\boldsymbol{x}_s)$:

$$\boldsymbol{x}_t = \mathrm{T}_{\boldsymbol{\theta}}(\boldsymbol{x}_s) := \frac{\sigma_t}{\sigma_s}\boldsymbol{x}_s + \sigma_t \mathrm{F}_{\boldsymbol{\theta}}(\boldsymbol{x}_s), \tag{33}$$

where $\mathrm{F}_{\boldsymbol{\theta}}(\boldsymbol{x}_s) := \int_{\kappa(s)}^{\kappa(t)} \boldsymbol{x\theta}(\boldsymbol{x}_{\phi(\tau)}, \phi(\tau)) \, d\tau$ accounts for the model's contribution, $\kappa(t) := \frac{\alpha_t}{\sigma_t}$ and its square represents the signal-to-noise ratio (SNR), and $\phi(\kappa(t)) := t$ denotes its inverse function.

## B. Further Details for the Iterative Operator (33)

By applying the variation-of-constants formula to Eq. (32), we obtain:

$$\boldsymbol{x}_t = e^{f_1(t)} \left( -\int_s^t e^{-f_1(r)} \frac{\alpha_r g^2(r)}{2\sigma_r^2} \boldsymbol{x_\theta}(\boldsymbol{x}_r, r) \, \mathrm{d}r + \boldsymbol{x}_s \right) \tag{34}$$

where $f_1(r) := \int_s^r f(z) + \frac{g^2(z)}{2\sigma_z^2} \mathrm{d}z$, $f(t) = \frac{\mathrm{d}\log\alpha_t}{\mathrm{d}t}$ and $g^2(t) = \frac{\mathrm{d}\sigma_t^2}{\mathrm{d}t} - 2\frac{\mathrm{d}\log\alpha_t}{\mathrm{d}t}\sigma_t^2$. Note that $f_1(r)$ can be rewritten as

$$f_1(r) = \int_s^r \frac{\mathrm{d}\log\alpha_\gamma}{\mathrm{d}\gamma} + \frac{1}{2\sigma_\gamma^2}\frac{\mathrm{d}\sigma_\gamma^2}{\mathrm{d}\gamma} - \frac{\mathrm{d}\log\alpha_\gamma}{\mathrm{d}\gamma}\mathrm{d}\gamma = \int_s^r \frac{1}{\sigma_\gamma}\frac{\mathrm{d}\sigma_\gamma}{\mathrm{d}\gamma}\mathrm{d}\gamma = \log\frac{\sigma_r}{\sigma_s}, \tag{35}$$

thus, $e^{f_1(t)} = \frac{\sigma_t}{\sigma_s}$ and $e^{-f_1(r)} = \frac{\sigma_s}{\sigma_r}$. Then, Eq. (34) can be rewritten as

$$\boldsymbol{x}_t = \frac{\sigma_t}{\sigma_s}\left( -\int_s^t \sigma_s \frac{\alpha_r}{\sigma_r}\frac{g^2(r)}{2\sigma_r^2}\boldsymbol{x_\theta}(\boldsymbol{x}_r, r)\,\mathrm{d}r + \boldsymbol{x}_s \right). \tag{36}$$

Note that $\frac{g^2(r)}{2\sigma_r^2} = \frac{1}{\sigma_r}\frac{\mathrm{d}\sigma_r}{\mathrm{d}r} - \frac{1}{\alpha_r}\frac{\mathrm{d}\alpha_r}{\mathrm{d}r}$, so $\frac{\alpha_r}{\sigma_r}\frac{g^2(r)}{2\sigma_r^2} = -\frac{\mathrm{d}}{\mathrm{d}r}\left(\frac{\alpha_r}{\sigma_r}\right)$. Therefore, Eq. (36) can be rewritten as:

$$\boldsymbol{x}_t = \frac{\sigma_t}{\sigma_s}\left( \int_s^t \sigma_s \boldsymbol{x_\theta}(\boldsymbol{x}_r, r)\frac{\mathrm{d}}{\mathrm{d}r}\left(\frac{\alpha_r}{\sigma_r}\right)\mathrm{d}r + \boldsymbol{x}_s \right) = \frac{\sigma_t}{\sigma_s}\boldsymbol{x}_s + \sigma_t\int_s^t \boldsymbol{x_\theta}(\boldsymbol{x}_r, r)\,\mathrm{d}\frac{\alpha_r}{\sigma_r}. \tag{37}$$

This result is equivalent to Eq. (33).

## C. The upper bound of $L_\mathrm{T}$

For any two inputs $\boldsymbol{x}_s^{(1)}, \boldsymbol{x}_s^{(2)}$, we have:

$$\mathrm{T}_{\boldsymbol{\theta}}(\boldsymbol{x}_s^{(1)}) - \mathrm{T}_{\boldsymbol{\theta}}(\boldsymbol{x}_s^{(2)}) = \frac{\sigma_t}{\sigma_s}(\boldsymbol{x}_s^{(1)} - \boldsymbol{x}_s^{(2)}) + \sigma_t(\kappa(t) - \kappa(s))[\boldsymbol{x_\theta}(\boldsymbol{x}_s^{(1)}, s) - \boldsymbol{x_\theta}(\boldsymbol{x}_s^{(2)}, s)] \tag{38}$$

From the parameterization in Eq. (31), we obtain:

$$\boldsymbol{x_\theta}(\boldsymbol{x}_s^{(1)}, s) - \boldsymbol{x_\theta}(\boldsymbol{x}_s^{(2)}, s) = \frac{1}{\alpha_s}[(\boldsymbol{x}_s^{(1)} - \boldsymbol{x}_s^{(2)}) - \sigma_s(\boldsymbol{\epsilon_\theta}(\boldsymbol{x}_s^{(1)}, s) - \boldsymbol{\epsilon_\theta}(\boldsymbol{x}_s^{(2)}, s))] \tag{39}$$

Assuming the noise prediction network $\boldsymbol{\epsilon_\theta}(\boldsymbol{x}, t)$ satisfies the Lipschitz condition with respect to $\boldsymbol{x}$:

$$\|\boldsymbol{\epsilon_\theta}(\boldsymbol{x}^{(1)}, t) - \boldsymbol{\epsilon_\theta}(\boldsymbol{x}^{(2)}, t)\| \le L_\epsilon\|\boldsymbol{x}^{(1)} - \boldsymbol{x}^{(2)}\|, \tag{40}$$

we can bound:

$$\|\boldsymbol{x_\theta}(\boldsymbol{x}_s^{(1)}, s) - \boldsymbol{x_\theta}(\boldsymbol{x}_s^{(2)}, s)\| \le \frac{1 + \sigma_s L_\epsilon}{\alpha_s}\|\boldsymbol{x}_s^{(1)} - \boldsymbol{x}_s^{(2)}\| \tag{41}$$

Combining these results, we obtain:

$$\|\mathrm{T}_{\boldsymbol{\theta}}(\boldsymbol{x}_s^{(1)}) - \mathrm{T}_{\boldsymbol{\theta}}(\boldsymbol{x}_s^{(2)})\| \le \frac{\sigma_t}{\sigma_s}\|\boldsymbol{x}_s^{(1)} - \boldsymbol{x}_s^{(2)}\| + \sigma_t(\kappa(t) - \kappa(s))\frac{1 + \sigma_s L_\epsilon}{\alpha_s}\|\boldsymbol{x}_s^{(1)} - \boldsymbol{x}_s^{(2)}\| \tag{42}$$

$$= \left[\frac{\sigma_t}{\sigma_s} + \sigma_t(\kappa(t) - \kappa(s))\frac{1 + \sigma_s L_\epsilon}{\alpha_s}\right]\|\boldsymbol{x}_s^{(1)} - \boldsymbol{x}_s^{(2)}\| \tag{43}$$

Recalling that $\kappa(t) = \frac{\alpha_t}{\sigma_t}$, we can simplify:

$$\sigma_t(\kappa(t) - \kappa(s))\frac{1 + \sigma_s L_\epsilon}{\alpha_s} = (1 + \sigma_s L_\epsilon)\left(\frac{\alpha_t}{\alpha_s} - \frac{\sigma_t}{\sigma_s}\right). \tag{44}$$

Therefore, an upper bound of the Lipschitz constant of the discretized iterative operator $\mathrm{T}_{\boldsymbol{\theta}}$ under first-order Euler approximation as follows:

$$L_\mathrm{T} \le \frac{\sigma_t}{\sigma_s} + (1 + \sigma_s L_\epsilon)\left(\frac{\alpha_t}{\alpha_s} - \frac{\sigma_t}{\sigma_s}\right). \tag{45}$$

This can be equivalently written as:

$$L_{\mathrm{T}} \leq (1 + \sigma_s L_\epsilon)\frac{\alpha_t}{\alpha_s} - \sigma_s L_\epsilon \frac{\sigma_t}{\sigma_s} \tag{46}$$

The Lipschitz constant $L_{\mathrm{T}}$ directly depends on the Lipschitz constant $L_\epsilon$ of the noise prediction network and the ratios of the diffusion scheduling parameters $\alpha_t/\alpha_s$ and $\sigma_t/\sigma_s$. When $L_{\mathrm{T}} < 1$, the operator $\mathrm{T}_{\boldsymbol{\theta}}$ is contractive, ensuring the stability and convergence of the iterative inference process.

### C.1. Complete Proof of Proposition 4.1

**Proposition C.1** (Loss of the sufficient upper-bound certificate). *The update* $\mathrm{T}_{\boldsymbol{\theta}}$ *violates contractivity when* $L_{\boldsymbol{x}_{\boldsymbol{\theta}}} \geq \frac{\sigma_s - \sigma_t}{\sigma_s \alpha_t - \sigma_t \alpha_s}$.

*Proof.* From the Lipschitz bound in Eq. (45):

$$L_T \leq \frac{\sigma_t}{\sigma_s} + \sigma_t h_t L_{\boldsymbol{x}_{\boldsymbol{\theta}}} \tag{47}$$

For contractivity ($L_T < 1$), we require:

$$\frac{\sigma_t}{\sigma_s} + \sigma_t \left( \frac{\alpha_t}{\sigma_t} - \frac{\alpha_s}{\sigma_s} \right) L_{\boldsymbol{x}_{\boldsymbol{\theta}}} < 1 \tag{48}$$

$$L_{\boldsymbol{x}_{\boldsymbol{\theta}}} < \frac{\sigma_s - \sigma_t}{\sigma_s \alpha_t - \sigma_t \alpha_s} \tag{49}$$

To show $\frac{1}{\alpha_t} > \frac{\sigma_s - \sigma_t}{\sigma_s \alpha_t - \sigma_t \alpha_s}$, cross-multiply:

$$\sigma_s \alpha_t - \sigma_t \alpha_s > \alpha_t \sigma_s - \alpha_t \sigma_t \quad \Leftrightarrow \quad \alpha_t > \alpha_s \tag{50}$$

This holds for denoising ($t < s$) since $\alpha_t$ increases as $t \to 0$.

**Special Cases:**

- **VP Schedule** ($\alpha_t^2 + \sigma_t^2 = 1$): The bound tightens as $t \to 0$.

- **EDM** ($\alpha_t \equiv 1$): The bound simplifies to $L_{\boldsymbol{x}_{\boldsymbol{\theta}}} < 1$.

The proof is complete. $\qquad\square$

## D. Complete Proofs for Theoretical Results

### D.1. Proof of Theorem 4.3

*Proof.* Let

$$\boldsymbol{u}_k = \boldsymbol{x}_k - \mathrm{T}_{\boldsymbol{\theta}}(\boldsymbol{x}_k) \tag{51}$$

denote the residual between the current state and the solver candidate. Then

$$(1 - \gamma_k)\boldsymbol{x}_k + \gamma_k \mathrm{T}_{\boldsymbol{\theta}}(\boldsymbol{x}_k) = \boldsymbol{x}_k - \gamma_k \boldsymbol{u}_k. \tag{52}$$

Define the clean-target error

$$\boldsymbol{e}_k = \boldsymbol{x}_k - \boldsymbol{x}^*. \tag{53}$$

The step-wise expected squared error can be written as

$$J(\gamma_k) = \mathbb{E}\left[ \|\boldsymbol{e}_k - \gamma_k \boldsymbol{u}_k\|^2 \right]. \tag{54}$$

Expanding the squared norm gives

$$\|\boldsymbol{e}_k - \gamma_k \boldsymbol{u}_k\|^2 = \|\boldsymbol{e}_k\|^2 - 2\gamma_k \langle \boldsymbol{u}_k, \boldsymbol{e}_k \rangle + \gamma_k^2 \|\boldsymbol{u}_k\|^2. \tag{55}$$

Taking expectations, we obtain

$$J(\gamma_k) = a_k - 2\gamma_k b_k + \gamma_k^2 c_k, \tag{56}$$

where

$$a_k = \mathbb{E}\left[\|\boldsymbol{x}_k - \boldsymbol{x}^*\|^2\right], \tag{57}$$

$$b_k = \mathbb{E}\left[\langle \boldsymbol{u}_k, \boldsymbol{x}_k - \boldsymbol{x}^*\rangle\right], \tag{58}$$

and

$$c_k = \mathbb{E}\left[\|\boldsymbol{u}_k\|^2\right]. \tag{59}$$

By assumption, $c_k > 0$. Hence, $J(\gamma_k)$ is a strictly convex quadratic function of $\gamma_k$. Its derivative is

$$\frac{dJ(\gamma_k)}{d\gamma_k} = -2b_k + 2c_k\gamma_k. \tag{60}$$

Setting the derivative to zero gives

$$\gamma_k^* = \frac{b_k}{c_k} = \frac{\mathbb{E}\left[\langle \boldsymbol{u}_k, \boldsymbol{x}_k - \boldsymbol{x}^*\rangle\right]}{\mathbb{E}\left[\|\boldsymbol{u}_k\|^2\right]}. \tag{61}$$

The proof is complete. □

## D.2. Proof of Theorem 4.4

*Proof.* Let

$$\boldsymbol{u}_k = \boldsymbol{x}_k - \mathrm{T}_{\boldsymbol{\theta}}(\boldsymbol{x}_k) \tag{62}$$

and

$$\boldsymbol{e}_k = \boldsymbol{x}_k - \boldsymbol{x}^*. \tag{63}$$

For any coefficient $\gamma_k$, the corrected update can be written as

$$(1 - \gamma_k)\boldsymbol{x}_k + \gamma_k\,\mathrm{T}_{\boldsymbol{\theta}}(\boldsymbol{x}_k) = \boldsymbol{x}_k - \gamma_k\boldsymbol{u}_k. \tag{64}$$

Thus, the step-wise expected squared error is

$$J(\gamma_k) = \mathbb{E}\left[\|\boldsymbol{e}_k - \gamma_k\boldsymbol{u}_k\|^2\right]. \tag{65}$$

As shown in the proof of Theorem 4.3,

$$J(\gamma_k) = a_k - 2\gamma_k b_k + \gamma_k^2 c_k, \tag{66}$$

where

$$a_k = \mathbb{E}\left[\|\boldsymbol{x}_k - \boldsymbol{x}^*\|^2\right], \tag{67}$$

$$b_k = \mathbb{E}\left[\langle \boldsymbol{u}_k, \boldsymbol{x}_k - \boldsymbol{x}^*\rangle\right], \tag{68}$$

and

$$c_k = \mathbb{E}\left[\|\boldsymbol{u}_k\|^2\right]. \tag{69}$$

The vanilla solver corresponds to $\gamma = 1$, because

$$\boldsymbol{x}_k - \boldsymbol{u}_k = \mathrm{T}_{\boldsymbol{\theta}}(\boldsymbol{x}_k). \tag{70}$$

By Theorem 4.3, the optimal coefficient is

$$\gamma_k^* = \frac{b_k}{c_k}. \tag{71}$$

Therefore,

$$J(\gamma_k^*) = a_k - 2\frac{b_k}{c_k}b_k + \left(\frac{b_k}{c_k}\right)^2 c_k$$

$$= a_k - \frac{b_k^2}{c_k}. \tag{72}$$

For the vanilla update,

$$J(1) = a_k - 2b_k + c_k. \tag{73}$$

Thus,

$$
\begin{aligned}
J(1) - J(\gamma_k^*) &= a_k - 2b_k + c_k - \left( a_k - \frac{b_k^2}{c_k} \right) \\
&= \frac{b_k^2}{c_k} - 2b_k + c_k \\
&= \frac{b_k^2 - 2b_k c_k + c_k^2}{c_k} \\
&= \frac{(b_k - c_k)^2}{c_k} \geq 0.
\end{aligned}
\tag{74}
$$

Hence,

$$J(\gamma_k^*) \leq J(1). \tag{75}$$

Equivalently,

$$\mathbb{E}\left[ \left\| \boldsymbol{x}_{k-1}^{\mathrm{Stein}} - \boldsymbol{x}^* \right\|^2 \right] \leq \mathbb{E}\left[ \left\| \mathrm{T}_{\boldsymbol{\theta}}(\boldsymbol{x}_k) - \boldsymbol{x}^* \right\|^2 \right]. \tag{76}$$

Equality holds if and only if

$$(b_k - c_k)^2 = 0, \tag{77}$$

that is,

$$b_k = c_k. \tag{78}$$

Equivalently, the vanilla coefficient $\gamma = 1$ already minimizes the step-wise objective. The proof is complete. $\square$

### D.3. Proof of Lemma 4.6

*Proof.* Let

$$\boldsymbol{x} \sim \mathcal{N}(\boldsymbol{\mu}, \sigma^2 \mathbf{I}) \tag{79}$$

with density

$$p(\boldsymbol{x}) = (2\pi\sigma^2)^{-d/2} \exp\left( -\frac{\|\boldsymbol{x} - \boldsymbol{\mu}\|^2}{2\sigma^2} \right). \tag{80}$$

Then

$$\nabla_{\boldsymbol{x}} p(\boldsymbol{x}) = -\frac{\boldsymbol{x} - \boldsymbol{\mu}}{\sigma^2} p(\boldsymbol{x}). \tag{81}$$

Equivalently,

$$(\boldsymbol{x} - \boldsymbol{\mu}) p(\boldsymbol{x}) = -\sigma^2 \nabla_{\boldsymbol{x}} p(\boldsymbol{x}). \tag{82}$$

Let $\boldsymbol{v} : \mathbb{R}^d \to \mathbb{R}^d$ be differentiable with suitable integrability and boundary decay. Then

$$\mathbb{E}\left[ \langle \boldsymbol{v}(\boldsymbol{x}), \boldsymbol{x} - \boldsymbol{\mu} \rangle \right] = \int \langle \boldsymbol{v}(\boldsymbol{x}), \boldsymbol{x} - \boldsymbol{\mu} \rangle \, p(\boldsymbol{x}) d\boldsymbol{x}. \tag{83}$$

Using the identity above,

$$\mathbb{E}\left[ \langle \boldsymbol{v}(\boldsymbol{x}), \boldsymbol{x} - \boldsymbol{\mu} \rangle \right] = -\sigma^2 \int \langle \boldsymbol{v}(\boldsymbol{x}), \nabla_{\boldsymbol{x}} p(\boldsymbol{x}) \rangle \, d\boldsymbol{x}. \tag{84}$$

By integration by parts,

$$-\int \langle \boldsymbol{v}(\boldsymbol{x}), \nabla_{\boldsymbol{x}} p(\boldsymbol{x}) \rangle \, d\boldsymbol{x} = \int (\nabla \cdot \boldsymbol{v})(\boldsymbol{x}) p(\boldsymbol{x}) d\boldsymbol{x}, \tag{85}$$

where the boundary term vanishes under the assumed integrability and decay conditions. Therefore,

$$\mathbb{E}\left[ \langle \boldsymbol{v}(\boldsymbol{x}), \boldsymbol{x} - \boldsymbol{\mu} \rangle \right] = \sigma^2 \mathbb{E}\left[ \nabla \cdot \boldsymbol{v}(\boldsymbol{x}) \right]. \tag{86}$$

The proof is complete. $\square$

## D.4. Proof of Theorem 4.7

*Proof.* From Theorem 4.3, the step-wise MSE-optimal coefficient is

$$\gamma_k^* = \frac{\mathbb{E}\left[\langle \boldsymbol{u}_k, \boldsymbol{x}_k - \boldsymbol{x}^* \rangle\right]}{\mathbb{E}\left[\|\boldsymbol{u}_k\|^2\right]}. \tag{87}$$

The denominator is computable from the solver residual. We focus on the numerator

$$N_k = \mathbb{E}\left[\langle \boldsymbol{u}_k, \boldsymbol{x}_k - \boldsymbol{x}^* \rangle\right]. \tag{88}$$

Expanding,

$$N_k = \mathbb{E}\left[\langle \boldsymbol{u}_k, \boldsymbol{x}_k \rangle\right] - \mathbb{E}\left[\langle \boldsymbol{u}_k, \boldsymbol{x}^* \rangle\right]. \tag{89}$$

Under the ideal forward noising coupling,

$$\boldsymbol{x}_k = \alpha_k \boldsymbol{x}^* + \sigma_k \boldsymbol{\epsilon}, \qquad \boldsymbol{\epsilon} \sim \mathcal{N}(\boldsymbol{0}, \mathbf{I}), \tag{90}$$

we have

$$\boldsymbol{x}_k | \boldsymbol{x}^* \sim \mathcal{N}\left(\alpha_k \boldsymbol{x}^*, \sigma_k^2 \mathbf{I}\right). \tag{91}$$

Thus,

$$\boldsymbol{x}^* = \frac{1}{\alpha_k} \boldsymbol{x}_k - \frac{1}{\alpha_k}\left(\boldsymbol{x}_k - \alpha_k \boldsymbol{x}^*\right). \tag{92}$$

Substituting this identity gives

$$\mathbb{E}\left[\langle \boldsymbol{u}_k, \boldsymbol{x}^* \rangle\right] = \frac{1}{\alpha_k} \mathbb{E}\left[\langle \boldsymbol{u}_k, \boldsymbol{x}_k \rangle\right] - \frac{1}{\alpha_k} \mathbb{E}\left[\langle \boldsymbol{u}_k, \boldsymbol{x}_k - \alpha_k \boldsymbol{x}^* \rangle\right]. \tag{93}$$

We now apply Lemma 4.6 conditionally on $\boldsymbol{x}^*$. Under this conditional Gaussian law, the mean is

$$\boldsymbol{\mu} = \alpha_k \boldsymbol{x}^* \tag{94}$$

and the vector field is

$$\boldsymbol{v}(\boldsymbol{x}_k) = \boldsymbol{u}_k. \tag{95}$$

Therefore,

$$\mathbb{E}\left[\langle \boldsymbol{u}_k, \boldsymbol{x}_k - \alpha_k \boldsymbol{x}^* \rangle \mid \boldsymbol{x}^*\right] = \sigma_k^2 \mathbb{E}\left[\nabla \cdot \boldsymbol{u}_k \mid \boldsymbol{x}^*\right]. \tag{96}$$

Taking expectation over $\boldsymbol{x}^*$, we obtain

$$\mathbb{E}\left[\langle \boldsymbol{u}_k, \boldsymbol{x}_k - \alpha_k \boldsymbol{x}^* \rangle\right] = \sigma_k^2 \mathbb{E}\left[\nabla \cdot \boldsymbol{u}_k\right]. \tag{97}$$

Hence,

$$\mathbb{E}\left[\langle \boldsymbol{u}_k, \boldsymbol{x}^* \rangle\right] = \frac{1}{\alpha_k} \mathbb{E}\left[\langle \boldsymbol{u}_k, \boldsymbol{x}_k \rangle\right] - \frac{\sigma_k^2}{\alpha_k} \mathbb{E}\left[\nabla \cdot \boldsymbol{u}_k\right]. \tag{98}$$

Substituting this expression into $N_k$, we get

$$N_k = \mathbb{E}\left[\langle \boldsymbol{u}_k, \boldsymbol{x}_k \rangle\right] - \frac{1}{\alpha_k} \mathbb{E}\left[\langle \boldsymbol{u}_k, \boldsymbol{x}_k \rangle\right] + \frac{\sigma_k^2}{\alpha_k} \mathbb{E}\left[\nabla \cdot \boldsymbol{u}_k\right]. \tag{99}$$

Thus,

$$N_k = \left(1 - \frac{1}{\alpha_k}\right) \mathbb{E}\left[\langle \boldsymbol{u}_k, \boldsymbol{x}_k \rangle\right] + \frac{\sigma_k^2}{\alpha_k} \mathbb{E}\left[\nabla \cdot \boldsymbol{u}_k\right]. \tag{100}$$

Finally,

$$\gamma_k^* = \frac{\left(1 - \frac{1}{\alpha_k}\right) \mathbb{E}\left[\langle \boldsymbol{u}_k, \boldsymbol{x}_k \rangle\right] + \frac{\sigma_k^2}{\alpha_k} \mathbb{E}\left[\nabla \cdot \boldsymbol{u}_k\right]}{\mathbb{E}\left[\|\boldsymbol{u}_k\|^2\right]}. \tag{101}$$

The proof is complete. □

### D.5. Proof of Theorem 4.8

*Proof.* Let

$$\boldsymbol{e}_k = \boldsymbol{x}_k - \boldsymbol{x}^* \tag{102}$$

denote the clean-target error. The exact SteinDiff update is

$$\boldsymbol{x}_{k-1}^{\text{Stein}} = \boldsymbol{x}_k - \gamma_k^* \boldsymbol{u}_k. \tag{103}$$

Therefore,

$$\boldsymbol{x}_{k-1}^{\text{Stein}} - \boldsymbol{x}^* = \boldsymbol{e}_k - \gamma_k^* \boldsymbol{u}_k. \tag{104}$$

For a general coefficient $\gamma$, define

$$J_k(\gamma) = \mathbb{E}\left[\|\boldsymbol{e}_k - \gamma \boldsymbol{u}_k\|^2\right]. \tag{105}$$

Expanding the squared norm gives

$$J_k(\gamma) = E_k - 2\gamma b_k + \gamma^2 c_k, \tag{106}$$

where

$$E_k = \mathbb{E}\left[\|\boldsymbol{e}_k\|^2\right], \tag{107}$$

$$b_k = \mathbb{E}\left[\langle \boldsymbol{u}_k, \boldsymbol{e}_k \rangle\right], \tag{108}$$

and

$$c_k = \mathbb{E}\left[\|\boldsymbol{u}_k\|^2\right]. \tag{109}$$

Substituting

$$\gamma_k^* = \frac{b_k}{c_k} \tag{110}$$

into $J_k(\gamma)$ gives

$$J_k(\gamma_k^*) = E_k - 2\frac{b_k}{c_k}b_k + \left(\frac{b_k}{c_k}\right)^2 c_k$$

$$= E_k - \frac{b_k^2}{c_k}. \tag{111}$$

Since

$$J_k(\gamma_k^*) = \mathbb{E}\left[\left\|\boldsymbol{x}_{k-1}^{\text{Stein}} - \boldsymbol{x}^*\right\|^2\right], \tag{112}$$

we obtain

$$\mathbb{E}\left[\left\|\boldsymbol{x}_{k-1}^{\text{Stein}} - \boldsymbol{x}^*\right\|^2\right] = E_k - \frac{b_k^2}{c_k}. \tag{113}$$

This proves the step-wise error decay identity.

If $E_k > 0$, define

$$\rho_k = \frac{b_k^2}{c_k E_k}. \tag{114}$$

Then

$$E_k - \frac{b_k^2}{c_k} = E_k\left(1 - \frac{b_k^2}{c_k E_k}\right)$$

$$= (1 - \rho_k)E_k. \tag{115}$$

Hence,

$$E_{k-1}^{\text{Stein}} = (1 - \rho_k)E_k. \tag{116}$$

We now show that $0 \leq \rho_k \leq 1$. Since $b_k^2 \geq 0$, $c_k > 0$, and $E_k > 0$, we have $\rho_k \geq 0$. For the upper bound, the Cauchy–Schwarz inequality gives

$$
\begin{aligned}
b_k^2 &= \left(\mathbb{E}\left[\langle \boldsymbol{u}_k, \boldsymbol{e}_k \rangle\right]\right)^2 \\
&\leq \mathbb{E}\left[\|\boldsymbol{u}_k\|^2\right] \mathbb{E}\left[\|\boldsymbol{e}_k\|^2\right] \\
&= c_k E_k.
\end{aligned}
\tag{117}
$$

Therefore,

$$
0 \leq \rho_k \leq 1.
\tag{118}
$$

Applying the one-step relation recursively along the exact SteinDiff trajectory gives

$$
E_0^{\text{Stein}} = E_N \prod_{k=1}^{N} (1 - \rho_k).
\tag{119}
$$

If there exists $\eta \in (0, 1]$ such that $\rho_k \geq \eta$ for all $k$, then

$$
1 - \rho_k \leq 1 - \eta.
\tag{120}
$$

Thus,

$$
E_0^{\text{Stein}} \leq (1 - \eta)^N E_N.
\tag{121}
$$

The proof is complete. $\square$

## D.6. Proof of Corollary 4.9

*Proof.* The vanilla solver candidate is

$$
\mathrm{T}_{\boldsymbol{\theta}}(\boldsymbol{x}_k) = \boldsymbol{x}_k - \boldsymbol{u}_k.
\tag{122}
$$

The exact SteinDiff update is

$$
\boldsymbol{x}_{k-1}^{\text{Stein}} = \boldsymbol{x}_k - \gamma_k^* \boldsymbol{u}_k.
\tag{123}
$$

Therefore, their difference is

$$
\begin{aligned}
\boldsymbol{x}_{k-1}^{\text{Stein}} - \mathrm{T}_{\boldsymbol{\theta}}(\boldsymbol{x}_k) &= (\boldsymbol{x}_k - \gamma_k^* \boldsymbol{u}_k) - (\boldsymbol{x}_k - \boldsymbol{u}_k) \\
&= (1 - \gamma_k^*) \boldsymbol{u}_k.
\end{aligned}
\tag{124}
$$

Since

$$
\gamma_k^* = \frac{b_k}{c_k},
\tag{125}
$$

we have

$$
1 - \gamma_k^* = \frac{c_k - b_k}{c_k}.
\tag{126}
$$

Thus,

$$
\begin{aligned}
\mathbb{E}\left[\|\boldsymbol{x}_{k-1}^{\text{Stein}} - \mathrm{T}_{\boldsymbol{\theta}}(\boldsymbol{x}_k)\|^2\right] &= \left(\frac{c_k - b_k}{c_k}\right)^2 \mathbb{E}\left[\|\boldsymbol{u}_k\|^2\right] \\
&= \frac{(c_k - b_k)^2}{c_k}.
\end{aligned}
\tag{127}
$$

Next, using

$$
\boldsymbol{x}_k - \boldsymbol{x}^* = \boldsymbol{u}_k + \mathrm{T}_{\boldsymbol{\theta}}(\boldsymbol{x}_k) - \boldsymbol{x}^*,
\tag{128}
$$

we take the inner product with $\boldsymbol{u}_k$ and then take expectation:

$$
\begin{aligned}
b_k &= \mathbb{E}\left[\langle \boldsymbol{u}_k, \boldsymbol{x}_k - \boldsymbol{x}^* \rangle\right] \\
&= \mathbb{E}\left[\|\boldsymbol{u}_k\|^2\right] + \mathbb{E}\left[\langle \boldsymbol{u}_k, \mathrm{T}_{\boldsymbol{\theta}}(\boldsymbol{x}_k) - \boldsymbol{x}^* \rangle\right] \\
&= c_k + \mathbb{E}\left[\langle \boldsymbol{u}_k, \mathrm{T}_{\boldsymbol{\theta}}(\boldsymbol{x}_k) - \boldsymbol{x}^* \rangle\right].
\end{aligned}
\tag{129}
$$

Hence,

$$c_k - b_k = -\mathbb{E}\left[\langle \boldsymbol{u}_k, \mathrm{T}_{\boldsymbol{\theta}}(\boldsymbol{x}_k) - \boldsymbol{x}^* \rangle\right]. \tag{130}$$

By the Cauchy–Schwarz inequality,

$$\begin{aligned}
(c_k - b_k)^2 &= \left(\mathbb{E}\left[\langle \boldsymbol{u}_k, \mathrm{T}_{\boldsymbol{\theta}}(\boldsymbol{x}_k) - \boldsymbol{x}^* \rangle\right]\right)^2 \\
&\leq \mathbb{E}\left[\|\boldsymbol{u}_k\|^2\right] \mathbb{E}\left[\|\mathrm{T}_{\boldsymbol{\theta}}(\boldsymbol{x}_k) - \boldsymbol{x}^*\|^2\right] \\
&= c_k \mathbb{E}\left[\|\mathrm{T}_{\boldsymbol{\theta}}(\boldsymbol{x}_k) - \boldsymbol{x}^*\|^2\right].
\end{aligned} \tag{131}$$

Substituting this bound into the previous expression yields

$$\mathbb{E}\left[\left\|\boldsymbol{x}_{k-1}^{\text{Stein}} - \mathrm{T}_{\boldsymbol{\theta}}(\boldsymbol{x}_k)\right\|^2\right] \leq \mathbb{E}\left[\|\mathrm{T}_{\boldsymbol{\theta}}(\boldsymbol{x}_k) - \boldsymbol{x}^*\|^2\right]. \tag{132}$$

Therefore, if the right-hand side tends to zero, then the SteinDiff update becomes asymptotically equivalent to the vanilla solver candidate in expected MSE.

The proof is complete. $\qquad\square$

### D.7. Proof of Theorem 4.10

*Proof.* Let

$$N_{p_k} = \left(1 - \frac{1}{\alpha_k}\right) \mathbb{E}_{p_k}\left[\langle \boldsymbol{u}_k, \boldsymbol{x}_k \rangle\right] + \frac{\sigma_k^2}{\alpha_k} \mathbb{E}_{p_k}\left[\nabla \cdot \boldsymbol{u}_k\right], \tag{133}$$

and

$$D_{p_k} = \mathbb{E}_{p_k}\left[\|\boldsymbol{u}_k\|^2\right]. \tag{134}$$

Similarly, define

$$N_{\tilde{p}_k} = \left(1 - \frac{1}{\alpha_k}\right) \mathbb{E}_{\tilde{p}_k}\left[\langle \boldsymbol{u}_k, \boldsymbol{x}_k \rangle\right] + \frac{\sigma_k^2}{\alpha_k} \mathbb{E}_{\tilde{p}_k}\left[\nabla \cdot \boldsymbol{u}_k\right], \tag{135}$$

and

$$D_{\tilde{p}_k} = \mathbb{E}_{\tilde{p}_k}\left[\|\boldsymbol{u}_k\|^2\right]. \tag{136}$$

Then

$$\gamma_k^* = \frac{N_{p_k}}{D_{p_k}}, \qquad \tilde{\gamma}_k = \frac{N_{\tilde{p}_k}}{D_{\tilde{p}_k}}. \tag{137}$$

By assumption, the denominators are bounded away from zero. Thus, there exists $\delta_k > 0$ such that

$$D_{p_k} \geq \delta_k, \qquad D_{\tilde{p}_k} \geq \delta_k. \tag{138}$$

By the expectation-shift assumption in Theorem 4.10, there exist finite constants $C_{N,k}$ and $C_{D,k}$ such that

$$|N_{\tilde{p}_k} - N_{p_k}| \leq C_{N,k} \mathcal{S}(\tilde{p}_k, p_k), \tag{139}$$

and

$$|D_{\tilde{p}_k} - D_{p_k}| \leq C_{D,k} \mathcal{S}(\tilde{p}_k, p_k). \tag{140}$$

We now bound

$$|\tilde{\gamma}_k - \gamma_k^*| = \left|\frac{N_{\tilde{p}_k}}{D_{\tilde{p}_k}} - \frac{N_{p_k}}{D_{p_k}}\right|. \tag{141}$$

Adding and subtracting $N_{p_k}/D_{\tilde{p}_k}$, we get

$$|\tilde{\gamma}_k - \gamma_k^*| \leq \left|\frac{N_{\tilde{p}_k} - N_{p_k}}{D_{\tilde{p}_k}}\right| + \left|N_{p_k}\left(\frac{1}{D_{\tilde{p}_k}} - \frac{1}{D_{p_k}}\right)\right|. \tag{142}$$

For the first term,

$$\left|\frac{N_{\tilde{p}_k} - N_{p_k}}{D_{\tilde{p}_k}}\right| \leq \frac{C_{N,k}}{\delta_k} \mathcal{S}(\tilde{p}_k, p_k). \tag{143}$$

For the second term,

$$\left| \frac{1}{D_{\tilde{p}_k}} - \frac{1}{D_{p_k}} \right| = \frac{|D_{p_k} - D_{\tilde{p}_k}|}{D_{\tilde{p}_k} D_{p_k}} \leq \frac{C_{D,k}}{\delta_k^2} \mathcal{S}(\tilde{p}_k, p_k). \tag{144}$$

Therefore,

$$\left| N_{p_k} \left( \frac{1}{D_{\tilde{p}_k}} - \frac{1}{D_{p_k}} \right) \right| \leq \frac{|N_{p_k}| C_{D,k}}{\delta_k^2} \mathcal{S}(\tilde{p}_k, p_k). \tag{145}$$

Combining the two bounds,

$$|\tilde{\gamma}_k - \gamma_k^*| \leq \left( \frac{C_{N,k}}{\delta_k} + \frac{|N_{p_k}| C_{D,k}}{\delta_k^2} \right) \mathcal{S}(\tilde{p}_k, p_k). \tag{146}$$

Define

$$C_k = \frac{C_{N,k}}{\delta_k} + \frac{|N_{p_k}| C_{D,k}}{\delta_k^2}. \tag{147}$$

Then

$$|\tilde{\gamma}_k - \gamma_k^*| \leq C_k \mathcal{S}(\tilde{p}_k, p_k). \tag{148}$$

This proves the perturbation bound.

For EDM-style parameterization, $\alpha_k \equiv 1$. Therefore,

$$1 - \frac{1}{\alpha_k} = 0, \tag{149}$$

and the drift-related term

$$\left( 1 - \frac{1}{\alpha_k} \right) \mathbb{E}\left[ \langle \boldsymbol{u}_k, \boldsymbol{x}_k \rangle \right] \tag{150}$$

vanishes from the numerator. The remaining numerator contains only the divergence-based term

$$\frac{\sigma_k^2}{\alpha_k} \mathbb{E}\left[ \nabla \cdot \boldsymbol{u}_k \right]. \tag{151}$$

The proof is complete. $\qquad\square$

## D.8. Proof of Theorem 4.11

*Proof.* The statement treats $\bar{\gamma}_k$ as a fixed coefficient, or equivalently conditions on its estimated value. Random finite-batch, Hutchinson, and clipping effects are treated as additional perturbations of the practical coefficient.

Recall that

$$J_k(\gamma) = \mathbb{E}_{p_k}\left[ \|(1-\gamma)\boldsymbol{x}_k + \gamma\, \mathrm{T}_{\boldsymbol{\theta}}(\boldsymbol{x}_k) - \boldsymbol{x}^*\|^2 \right]. \tag{152}$$

Using

$$\boldsymbol{u}_k = \boldsymbol{x}_k - \mathrm{T}_{\boldsymbol{\theta}}(\boldsymbol{x}_k), \tag{153}$$

we have

$$(1-\gamma)\boldsymbol{x}_k + \gamma\, \mathrm{T}_{\boldsymbol{\theta}}(\boldsymbol{x}_k) = \boldsymbol{x}_k - \gamma\boldsymbol{u}_k. \tag{154}$$

Therefore,

$$J_k(\gamma) = \mathbb{E}_{p_k}\left[ \|\boldsymbol{x}_k - \boldsymbol{x}^* - \gamma\boldsymbol{u}_k\|^2 \right]. \tag{155}$$

As above, this is the quadratic function

$$J_k(\gamma) = a_k - 2b_k\gamma + c_k\gamma^2, \tag{156}$$

where

$$a_k = \mathbb{E}_{p_k}\left[ \|\boldsymbol{x}_k - \boldsymbol{x}^*\|^2 \right], \tag{157}$$

$$b_k = \mathbb{E}_{p_k}\left[ \langle \boldsymbol{u}_k, \boldsymbol{x}_k - \boldsymbol{x}^* \rangle \right], \tag{158}$$

and

$$c_k = \mathbb{E}_{p_k}\left[ \|\boldsymbol{u}_k\|^2 \right]. \tag{159}$$

The minimizer is

$$\gamma_k^* = \frac{b_k}{c_k}. \tag{160}$$

Completing the square,

$$J_k(\gamma) = a_k - \frac{b_k^2}{c_k} + c_k \left(\gamma - \frac{b_k}{c_k}\right)^2. \tag{161}$$

Since

$$J_k(\gamma_k^*) = a_k - \frac{b_k^2}{c_k}, \tag{162}$$

we obtain

$$J_k(\gamma) = J_k(\gamma_k^*) + c_k \left(\gamma - \gamma_k^*\right)^2. \tag{163}$$

Setting $\gamma = \bar{\gamma}_k$, we get

$$J_k(\bar{\gamma}_k) = J_k(\gamma_k^*) + c_k \left(\bar{\gamma}_k - \gamma_k^*\right)^2. \tag{164}$$

This proves the first claim.

The correction with $\bar{\gamma}_k$ improves over the vanilla update if

$$J_k(\bar{\gamma}_k) \le J_k(1). \tag{165}$$

Using the identity above, this condition becomes

$$J_k(\gamma_k^*) + c_k \left(\bar{\gamma}_k - \gamma_k^*\right)^2 \le J_k(1). \tag{166}$$

Rearranging yields

$$c_k \left(\bar{\gamma}_k - \gamma_k^*\right)^2 \le J_k(1) - J_k(\gamma_k^*). \tag{167}$$

The proof is complete. $\qquad\square$

### D.9. Proof of Corollary 4.12

*Proof.* From Theorem 4.11, for any fixed coefficient $\bar{\gamma}_k$,

$$J_k(\bar{\gamma}_k) - J_k(\gamma_k^*) = c_k \left(\bar{\gamma}_k - \gamma_k^*\right)^2. \tag{168}$$

Choose

$$\bar{\gamma}_k = \tilde{\gamma}_k. \tag{169}$$

Then

$$J_k(\tilde{\gamma}_k) - J_k(\gamma_k^*) = c_k \left(\tilde{\gamma}_k - \gamma_k^*\right)^2. \tag{170}$$

By Theorem 4.10,

$$|\tilde{\gamma}_k - \gamma_k^*| \le C_k \mathcal{S}(\tilde{p}_k, p_k). \tag{171}$$

Squaring both sides gives

$$(\tilde{\gamma}_k - \gamma_k^*)^2 \le C_k^2 \mathcal{S}(\tilde{p}_k, p_k)^2. \tag{172}$$

Multiplying by $c_k$, we obtain

$$J_k(\tilde{\gamma}_k) - J_k(\gamma_k^*) \le c_k C_k^2 \mathcal{S}(\tilde{p}_k, p_k)^2. \tag{173}$$

This proves the controlled degradation bound.

Furthermore, if

$$c_k C_k^2 \mathcal{S}(\tilde{p}_k, p_k)^2 \le J_k(1) - J_k(\gamma_k^*), \tag{174}$$

then

$$J_k(\tilde{\gamma}_k) \le J_k(1). \tag{175}$$

Thus, when the score deviation is small relative to the optimality gap, the distribution-shifted correction preserves the step-wise improvement over the vanilla update.

The proof is complete. $\qquad\square$

### D.10. EDM Simplification

**Proposition D.1** (EDM simplification of the correction coefficient). *Under EDM-style parameterization with $\alpha_k \equiv 1$, the reference-free coefficient in Eq. (11) reduces to*

$$\gamma^*_{k,\text{EDM}} = \frac{\sigma_k^2 \mathbb{E}\left[\nabla \cdot \boldsymbol{u}_k\right]}{\mathbb{E}\left[\|\boldsymbol{u}_k\|^2\right]}. \tag{176}$$

*Proof.* Starting from Eq. (11),

$$\gamma^*_k = \frac{\left(1 - \frac{1}{\alpha_k}\right)\mathbb{E}\left[\langle \boldsymbol{u}_k, \boldsymbol{x}_k\rangle\right] + \frac{\sigma_k^2}{\alpha_k}\mathbb{E}\left[\nabla \cdot \boldsymbol{u}_k\right]}{\mathbb{E}\left[\|\boldsymbol{u}_k\|^2\right]}. \tag{177}$$

For EDM-style parameterization, $\alpha_k \equiv 1$. Therefore,

$$1 - \frac{1}{\alpha_k} = 0, \qquad \frac{\sigma_k^2}{\alpha_k} = \sigma_k^2. \tag{178}$$

Substituting these identities gives

$$\gamma^*_{k,\text{EDM}} = \frac{\sigma_k^2 \mathbb{E}\left[\nabla \cdot \boldsymbol{u}_k\right]}{\mathbb{E}\left[\|\boldsymbol{u}_k\|^2\right]}. \tag{179}$$

The proof is complete. $\square$

**Corollary D.2** (EDM Stability via Geometric Decoupling). *The EDM parameterization achieves inherent stability through:*

1. ***Removal of $\alpha_t$-induced singularities***: *In VP schedules, as $\alpha_t \to 0$ (high noise), the term $1/\alpha_k$ can become numerically unstable. EDM avoids this entirely.*

2. ***Pure geometric signal***: *The correction only depends on local manifold geometry (divergence), not on global data scaling.*

3. ***Simplified estimation***: *Fewer terms to estimate reduces variance in the Hutchinson estimator.*

*Proof.* For VP schedules with $\alpha_t^2 + \sigma_t^2 = 1$:

- At high noise levels ($t$ large): $\alpha_t \to 0$, causing $(1 - 1/\alpha_t) \to -\infty$.

- The drift term magnitude $|1 - 1/\alpha_t| \cdot |\mathbb{E}[\langle \boldsymbol{u}_k, \boldsymbol{x}_k\rangle]|$ can dominate and destabilize the correction.

In contrast, EDM's $\alpha_t = 1$ yields $(1 - 1/\alpha_t) = 0$ uniformly across all noise levels. $\square$

**Remark D.3.** *This proposition only states an algebraic simplification of the SteinDiff coefficient under $\alpha_k \equiv 1$. It does not by itself imply unconditional stability or improved robustness of EDM-style parameterization.*

### D.11. Self-Consistency Correction of SteinDiff

This section describes an optional self-consistency correction for SteinDiff. The goal is to compare the one-step SteinDiff proposal with a trajectory-based look-ahead estimate and blend the two when they are consistent. This variant is intended as an empirical enhancement for aggressive discretization regimes and is not part of the core theoretical guarantees in the main text.

**Proposition D.4** (Log-sigma interpolation weight). *Let $\lambda_t = \log \sigma_t$. If the intermediate trajectory is approximated by linear interpolation in the $\lambda$-coordinate between $t_k$ and $t_{k-2}$, then the interpolation weight for estimating the state at $t_{k-1}$ is*

$$w = \frac{\log(\sigma_{t_{k-1}}/\sigma_{t_k})}{\log(\sigma_{t_{k-2}}/\sigma_{t_k})}. \tag{184}$$

**Algorithm 2** Optional SteinDiff with Trajectory Self-Consistency

**Require:** Model $\boldsymbol{x_\theta}$; update operator $\mathrm{T_\theta}$; schedule $\{t_i, \alpha_{t_i}, \sigma_{t_i}\}$; batch size $B$.

1: Initialize $\{\boldsymbol{x}_{t_k}^{(i)}, t_k, t_{k-1}, t_{k-2}\}_{i=1}^B$.

2: $(\boldsymbol{x}_p^{(i)}, \hat{\gamma}_k) \leftarrow \mathrm{SteinDiff}(\boldsymbol{x}_{t_k}^{(i)}, t_k, t_{k-1}, \boldsymbol{x_\theta}, \{\alpha, \sigma\})$.

3: $\boldsymbol{x}_{two}^{(i)} \leftarrow \mathrm{TwoStep}(\boldsymbol{x}_{t_k}^{(i)}, t_k, t_{k-2})$.

4:
$$w = \frac{\log(\sigma_{t_{k-1}}/\sigma_{t_k})}{\log(\sigma_{t_{k-2}}/\sigma_{t_k})}. \tag{180}$$

5:
$$\boldsymbol{x}_{alt}^{(i)} \leftarrow (1-w)\boldsymbol{x}_{t_k}^{(i)} + w\boldsymbol{x}_{two}^{(i)}. \tag{181}$$

6:
$$\rho_k^{(i)} \leftarrow \exp\left(-\frac{\|\boldsymbol{x}_p^{(i)} - \boldsymbol{x}_{alt}^{(i)}\|^2}{2\sigma_{t_{k-1}}^2}\right). \tag{182}$$

7:
$$\boldsymbol{x}_{t_{k-1}}^{(i)} \leftarrow \rho_k^{(i)}\boldsymbol{x}_p^{(i)} + (1 - \rho_k^{(i)})\boldsymbol{x}_{alt}^{(i)}. \tag{183}$$

8: **return** $\{\boldsymbol{x}_{t_{k-1}}^{(i)}\}_{i=1}^B$.

*Proof.* Under the log-$\sigma$ coordinate $\lambda_t = \log \sigma_t$, linear interpolation gives

$$\boldsymbol{x}(\lambda_{t_{k-1}}) \approx (1-w)\boldsymbol{x}(\lambda_{t_k}) + w\boldsymbol{x}(\lambda_{t_{k-2}}), \tag{185}$$

where

$$w = \frac{\lambda_{t_{k-1}} - \lambda_{t_k}}{\lambda_{t_{k-2}} - \lambda_{t_k}}. \tag{186}$$

Substituting $\lambda_t = \log \sigma_t$ yields

$$w = \frac{\log \sigma_{t_{k-1}} - \log \sigma_{t_k}}{\log \sigma_{t_{k-2}} - \log \sigma_{t_k}} = \frac{\log(\sigma_{t_{k-1}}/\sigma_{t_k})}{\log(\sigma_{t_{k-2}}/\sigma_{t_k})}. \tag{187}$$

The proof is complete. $\square$

**Proposition D.5** (Gaussian compatibility weight). *Assume that the discrepancy between the SteinDiff proposal $\boldsymbol{x}_p$ and the trajectory-based estimate $\boldsymbol{x}_{alt}$ follows an isotropic Gaussian compatibility model with scale $\tau_k$. Then*

$$\rho_k = \exp\left(-\frac{\|\boldsymbol{x}_p - \boldsymbol{x}_{alt}\|^2}{2\tau_k^2}\right) \tag{188}$$

*is the normalized likelihood of the discrepancy relative to perfect agreement. In Algorithm 2, we use $\tau_k = \sigma_{t_{k-1}}$.*

*Proof.* Consider the compatibility model

$$\boldsymbol{x}_p \mid \boldsymbol{x}_{alt} \sim \mathcal{N}\left(\boldsymbol{x}_{alt}, \tau_k^2\mathbf{I}\right). \tag{189}$$

The likelihood is proportional to

$$p(\boldsymbol{x}_p \mid \boldsymbol{x}_{alt}) \propto \exp\left(-\frac{\|\boldsymbol{x}_p - \boldsymbol{x}_{alt}\|^2}{2\tau_k^2}\right). \tag{190}$$

The likelihood at perfect agreement, $\boldsymbol{x}_p = \boldsymbol{x}_{alt}$, is the maximum value. Normalizing by this maximum gives

$$\rho_k = \exp\left(-\frac{\|\boldsymbol{x}_p - \boldsymbol{x}_{alt}\|^2}{2\tau_k^2}\right). \tag{191}$$

Thus, $\rho_k \in (0, 1]$, with $\rho_k = 1$ under perfect agreement and smaller values when the two estimates disagree.

The proof is complete. $\square$

**Proposition D.6** (Perturbation induced by self-consistency blending). *Let*

$$\boldsymbol{x}^{SC} = \rho \boldsymbol{x}_p + (1 - \rho) \boldsymbol{x}_{alt}, \qquad 0 \le \rho \le 1. \tag{192}$$

*Then the deviation from the SteinDiff proposal is*

$$\boldsymbol{x}^{SC} - \boldsymbol{x}_p = (1 - \rho)(\boldsymbol{x}_{alt} - \boldsymbol{x}_p), \tag{193}$$

*and therefore*

$$\|\boldsymbol{x}^{SC} - \boldsymbol{x}_p\| \le \|\boldsymbol{x}_{alt} - \boldsymbol{x}_p\|. \tag{194}$$

*Moreover, for any target $\boldsymbol{x}^*$,*

$$\|\boldsymbol{x}^{SC} - \boldsymbol{x}^*\|^2 = \|\boldsymbol{x}_p - \boldsymbol{x}^*\|^2 + 2(1 - \rho)\langle \boldsymbol{x}_p - \boldsymbol{x}^*, \boldsymbol{x}_{alt} - \boldsymbol{x}_p \rangle + (1 - \rho)^2 \|\boldsymbol{x}_{alt} - \boldsymbol{x}_p\|^2. \tag{195}$$

*Consequently, any improvement or degradation relative to the SteinDiff proposal is controlled by the discrepancy $\boldsymbol{x}_{alt} - \boldsymbol{x}_p$ and its alignment with the SteinDiff error.*

*Proof.* The first identity follows directly from

$$\boldsymbol{x}^{SC} = \rho \boldsymbol{x}_p + (1 - \rho) \boldsymbol{x}_{alt} = \boldsymbol{x}_p + (1 - \rho)(\boldsymbol{x}_{alt} - \boldsymbol{x}_p). \tag{196}$$

Since $0 \le \rho \le 1$,

$$\|\boldsymbol{x}^{SC} - \boldsymbol{x}_p\| = (1 - \rho)\|\boldsymbol{x}_{alt} - \boldsymbol{x}_p\| \le \|\boldsymbol{x}_{alt} - \boldsymbol{x}_p\|. \tag{197}$$

For the target-error identity, write

$$\boldsymbol{x}^{SC} - \boldsymbol{x}^* = \boldsymbol{x}_p - \boldsymbol{x}^* + (1 - \rho)(\boldsymbol{x}_{alt} - \boldsymbol{x}_p). \tag{198}$$

Expanding the squared norm gives the stated equality. □

**Remark D.7.** *The self-consistency correction is an optional empirical variant. The propositions above justify the interpolation weight and the compatibility weight, and show that the blended update stays close to the SteinDiff proposal when the two estimates agree. They do not imply an unconditional reduction of the clean-target MSE relative to the core SteinDiff update.*

## E. Experimental Setup and Details

Our experiments were conducted to validate the effectiveness of SteinDiff in enhancing state-of-the-art ODE solvers, including DPM-Solver++, UniPC, and the Heun method, using their default public implementations. We performed evaluations on several standard benchmarks with corresponding pre-trained models: a CIFAR-10 EDM model, an ImageNet 64x64 model under both EDM and logSNR noise schedules, and a Latent Diffusion Model for LSUN Bedrooms 256x256. Specifically, experiments on CIFAR-10 and ImageNet were conducted using an NVIDIA 3090 GPU, while evaluations on the LSUN Bedrooms 256x256 dataset utilized an NVIDIA 4090D GPU. Performance was measured using standard metrics, including Fréchet Inception Distance (FID) and Inception Score (IS), with FID scores computed over 50,000 generated samples. We also reported FD-DINOv2, which replaces the standard InceptionV3 encoder with a DINOv2 backbone for a more perceptually aligned evaluation. The SteinDiff regularizer was implemented as detailed in Algorithm 1, utilizing the Hutchinson trace estimator with five random vectors to approximate the divergence term. For the main quantitative results, we employed a fast KM variant to ensure accelerated convergence while maintaining theoretical guarantees. A small safeguard constant, $\epsilon$, was used to ensure numerical stability during the computation of the stabilization parameter $\hat{\gamma}_k$.

*Efficient Implementation.* The main computational cost lies in evaluating the divergence term $\nabla \cdot \boldsymbol{u}_k$ in Eq. (11), which naively requires $\mathcal{O}(d)$ vector-Jacobian products. We address this using the Hutchinson trace estimator (Hutchinson, 1989). This reduces the computational cost to a single vector-Jacobian product per sample. The estimator is unbiased and has been widely adopted in modern deep learning applications (Grathwohl et al., 2019; Tsitsulin et al., 2020; Meyer et al., 2021).

*Computational Overhead.* The cost of estimating the divergence for a batch of size $B$ using $m$ Hutchinson probes corresponds to $m$ vector-Jacobian products (VJPs). Crucially, these $m$ VJPs are highly *parallelizable* across GPU hardware, effectively minimizing the wall-clock latency compared to the primary neural network evaluations in $\mathrm{T}_{\boldsymbol{\theta}}$. Furthermore, SteinDiff can

be deployed *adaptively* and is activated only at critical timesteps where geometric stabilization is most required, rather than at every step of the trajectory. Due to this sparse activation, the amortized computational cost remains significantly lower than the cumulative overhead of increasing solver steps (NFE). Consequently, SteinDiff circumvents the need for additional NFEs, model retraining, or expensive schedule optimization, thereby offering a superior trade-off between efficiency and stability for high-quality generative inference.

### E.0.1. FORMAL COMPUTATIONAL COMPLEXITY ANALYSIS

**Proposition E.1** (Computational Overhead). *The per-step computational cost of SteinDiff, relative to a baseline ODE solver, is:*

$$Cost_{SteinDiff} = Cost_{Baseline} + O(m \cdot VJP) \tag{199}$$

*where $m$ is the number of Hutchinson probes and VJP denotes a vector-Jacobian product.*

*Proof.* At each step, SteinDiff requires:

1. Compute $\boldsymbol{u}_k = \boldsymbol{x}_k - \mathrm{T}_{\boldsymbol{\theta}}(\boldsymbol{x}_k)$: Already computed by baseline (free).

2. Compute $\hat{s}_{xu} = \frac{1}{B}\sum_{i=1}^{B}\langle \boldsymbol{u}_k^{(i)}, \boldsymbol{x}_k^{(i)}\rangle$: $O(Bd)$ operations.

3. Compute $\hat{s}_{uu} = \frac{1}{B}\sum_{i=1}^{B}\|\boldsymbol{u}_k^{(i)}\|^2$: $O(Bd)$ operations.

4. Compute divergence via Hutchinson: $m$ VJP calls.

5. Compute $\hat{\gamma}_k$ and update: $O(1)$ operations.

The VJP computation dominates. Each VJP has complexity comparable to one forward pass through the Jacobian. For neural networks, this is $O(\text{params})$ via backpropagation. The $m$ VJP calls are embarrassingly parallel across the batch dimension on modern GPUs. $\square$

*Table 2.* Comparison of computational overhead across methods.

| Method | Extra NFE | Extra VJP | Retraining |
|---|---|---|---|
| Baseline (DPM-Solver++) | 0 | 0 | No |
| SteinDiff (ours) | 0 | $m$ per step (parallelizable) | No |
| DPM-Solver-v3 | 0 | 0 | Reference solution |
| Restart Sampling | +50-100% | 0 | No |
| Consistency Distillation | 0 | 0 | Yes (expensive) |

**Remark E.2** (Adaptive Strategy & Amortized Efficiency). *When SteinDiff is applied adaptively (only at critical timesteps), the total overhead satisfies:*

$$Total\ Extra\ Cost \le K \cdot m \cdot VJP \tag{200}$$

*where $K \ll N$ is the number of critical steps (typically $K \le 3$ for early/high-noise stages). Crucially, since the $m$ VJP evaluations are parallelizable across the batch dimension, the impact on inference latency is minimal.*

### E.1. Details

We implement `SteinDiff` as a standalone PyTorch module compatible with standard ODE solvers (e.g., `DPM_Solver`). The design strictly adheres to the theoretical derivations in Section 4, enforcing a post-hoc correction step to stabilize the sampling trajectory. The stabilization mechanism acts via the `apply` method. Rather than accepting the raw solver output `x_t` as the next state, we enforce a KM formulation update using adaptive interpolation:

$$x_{\text{new}} = (1 - \gamma) \cdot x + \gamma \cdot x_t \tag{201}$$

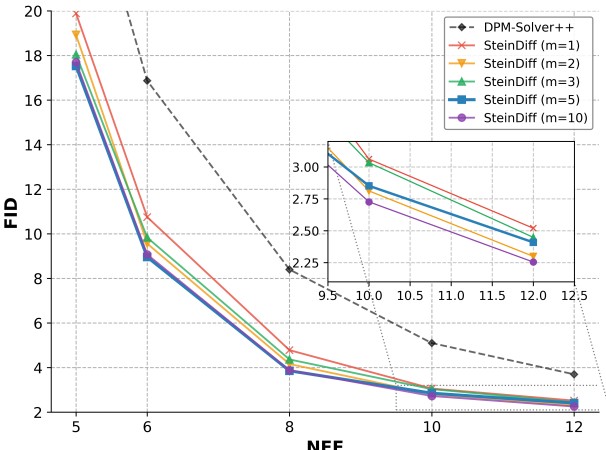

*Figure 9.* Sensitivity analysis of the Hutchinson probe count ($m$) on CIFAR-10. We compare SteinDiff with varying probe counts $m \in \{1, 2, 3, 5, 10\}$ against the DPM-Solver++ baseline. The results demonstrate that SteinDiff is highly robust to estimation noise, significantly outperforming the baseline even with a single probe ($m = 1$), and performance saturates rapidly at $m = 5$.

Here, x is the current state, x_t is the candidate prediction from the ODE solver, and $\gamma$ is the adaptive stabilization parameter computed dynamically at each step. The module intercepts the solver's candidate state and the original state to output the final, stabilized result.

The numerical evaluation of the optimal stabilization parameter $\gamma$, following the closed-form derivation in Eq. (11), is implemented within the compute_optimal_gamma method. The process begins by defining the update residual $u = x - x_t$ and aggregating the batch-wise statistics, specifically the inner product $\langle u, x \rangle$ (s_xu) and the squared norm $\|u\|^2$ (s_uu). A critical step involves estimating the divergence term $\nabla \cdot u$. To this end, we provide two strategies: an analytical approximation, which efficiently computes divergence solely on the linear component of the ODE operator but ignores non-linearities; and a full divergence estimator, which utilizes the Hutchinson trace method to capture the full residual divergence. The latter employs torch.autograd.functional.jvp combined with antithetic sampling and Rademacher distributions for variance reduction. Finally, these statistics are combined via Stein's Identity to solve for $\gamma$, with the result clamped to a lower bound (typically 1e-6) to satisfy the structure of stabilization.

To balance theoretical precision with computational cost, the module supports three configuration modes controlled via initialization flags. The full mode estimates the full residual divergence (including non-linearities) via the Hutchinson trace estimator at every step, maximizing theoretical rigor at the cost of additional overhead. Conversely, the analytical-only mode relies exclusively on the linear approximation of the ODE operator, adding minimal cost to the inference process. To bridge these extremes, we design an adaptive mode that dynamically switches between estimators based on the noise level $\sigma_t$. This hybrid approach utilizes the fast analytical approximation during early, high-noise stages and transitions to the precise Hutchinson estimator in the critical low-noise regime where fine-grained stability is paramount.

### E.2. More Experiments

We present further qualitative results demonstrating SteinDiff's robustness across three critical regimes. Figure 10 illustrates the severity of the contractivity trap at a strict budget of 5 NFE. Under these conditions, standard solvers (UniPC, DPM-Solver++) suffer geometric collapse and significant artifacts. SteinDiff (+DPM) effectively counters this drift by regularizing updates towards the data manifold, preserving coherent object structures across diverse classes.

Even when baselines converge at higher steps (10 NFE), geometric stabilization yields tangible perceptual benefits. As shown in Figure 11, while DPM-Solver++ produces globally coherent samples, it often smooths over textures. SteinDiff recovers fine-grained details specifically by sharpening features such as animal fur and flower petals, which forces the generation trajectory to adhere more closely to the underlying data geometry.

Regarding scalability, Figure 12 validates our Hutchinson-based trace estimation in high-dimensional spaces. On the LSUN-Bedrooms $256 \times 256$ benchmark (LDM), SteinDiff achieves a SOTA FID of 2.77 at 20 NFE. These results confirm that the proposed approximation remains computationally effective and robust in latent spaces.

Finally, the sensitivity analysis in Figure 9 reveals high robustness to the Hutchinson probe count $m$. Even a single probe ($m = 1$) yields significant gains over the baseline, and performance saturates rapidly at $m = 5$, indicating that the scalar stabilization parameter is empirically insensitive to gradient-estimation noise in our tested settings.

*Table 3.* Comprehensive performance comparison of various samplers on the imagenet 64×64 dataset with and without SteinDiff. The table contrasts the baseline performance (Base) of DPM-Solver++, UniPC, and Heun against the performance with fast SteinDiff (+Stein) under both logSNR and EDM schedules. Performance is evaluated using FID and FD-DINOv2 metrics, where lower scores are better. The percentage improvement (Improv. %) highlights the consistent and significant gains achieved by SteinDiff across all configurations.

| Metric | Schedule | Steps | DPM-Solver++ | | | UniPC | | | Heun | | |
|---|---|---|---|---|---|---|---|---|---|---|---|
| | | | Base | Stein | Improv. (%) | Base | Stein | Improv. (%) | Base | Stein | Improv. (%) |
| FID | logSNR | 3 | 20.92 | 16.48 | **21.2%** | 27.70 | 19.92 | **28.1%** | 230.05 | 124.69 | **45.8%** |
| | | 3.5 | 12.81 | 10.43 | **18.6%** | 17.04 | 13.05 | **23.4%** | 153.27 | 89.66 | **41.5%** |
| | | 4.5 | 6.16 | 5.22 | **15.3%** | 6.44 | 5.27 | **18.2%** | 51.25 | 33.47 | **34.7%** |
| | | 5.5 | 3.90 | 3.43 | **12.0%** | 3.32 | 2.87 | **13.6%** | 18.60 | 13.40 | **27.9%** |
| | | 6.5 | 2.96 | 2.67 | **9.7%** | 2.45 | 2.21 | **9.8%** | 8.49 | 6.81 | **19.8%** |
| | | 8 | 2.33 | 2.16 | **7.4%** | 2.02 | 1.88 | **7.1%** | 5.12 | 4.42 | **13.6%** |
| | | 10.5 | 1.96 | 1.86 | **5.1%** | 1.80 | 1.71 | **4.9%** | 2.43 | 2.24 | **7.8%** |
| | | 13 | 1.83 | 1.76 | **3.6%** | 1.73 | 1.68 | **2.9%** | 2.08 | 2.02 | **2.7%** |
| FD-DINOv2 | logSNR | 3 | 311.03 | 251.00 | **19.3%** | 425.71 | 315.45 | **25.9%** | 1923.78 | 1035.00 | **46.2%** |
| | | 3.5 | 232.80 | 193.00 | **17.1%** | 331.58 | 258.20 | **22.1%** | 1355.47 | 788.88 | **41.8%** |
| | | 4.5 | 160.59 | 137.79 | **14.2%** | 193.84 | 160.31 | **17.3%** | 697.94 | 452.96 | **35.1%** |
| | | 5.5 | 131.78 | 116.63 | **11.5%** | 138.25 | 120.55 | **12.8%** | 351.30 | 251.88 | **28.3%** |
| | | 6.5 | 118.60 | 107.59 | **9.3%** | 118.69 | 107.90 | **9.1%** | 231.57 | 184.33 | **20.4%** |
| | | 8 | 108.51 | 100.70 | **7.2%** | 106.82 | 99.45 | **6.9%** | 176.95 | 152.02 | **14.1%** |
| | | 10.5 | 101.62 | 95.73 | **5.8%** | 99.91 | 94.71 | **5.2%** | 120.77 | 110.50 | **8.5%** |
| | | 13 | 98.67 | 94.23 | **4.5%** | 97.15 | 93.46 | **3.8%** | 109.78 | 105.17 | **4.2%** |
| FID | EDM | 3 | 17.95 | 14.84 | **17.3%** | 30.02 | 22.58 | **24.8%** | 233.29 | 137.17 | **41.2%** |
| | | 3.5 | 11.00 | 9.45 | **14.1%** | 17.61 | 14.10 | **19.9%** | 81.06 | 50.66 | **37.5%** |
| | | 4.5 | 5.48 | 4.86 | **11.2%** | 7.37 | 6.32 | **14.2%** | 28.66 | 18.91 | **34.0%** |
| | | 5.5 | 3.66 | 3.33 | **9.0%** | 4.37 | 3.88 | **11.3%** | 10.78 | 8.07 | **25.1%** |
| | | 6.5 | 2.87 | 2.67 | **6.9%** | 3.33 | 3.05 | **8.4%** | 5.57 | 4.60 | **17.4%** |
| | | 8 | 2.33 | 2.20 | **5.5%** | 2.59 | 2.42 | **6.5%** | 3.56 | 3.12 | **12.4%** |
| | | 10.5 | 1.97 | 1.89 | **4.1%** | 2.02 | 1.92 | **4.8%** | 2.19 | 2.02 | **7.9%** |
| | | 13 | 1.83 | 1.77 | **3.2%** | 1.81 | 1.74 | **3.8%** | 1.94 | 1.87 | **3.4%** |
| FD-DINOv2 | EDM | 3 | 273.01 | 225.99 | **17.2%** | 449.23 | 335.90 | **25.2%** | 2010.42 | 1178.11 | **41.4%** |
| | | 3.5 | 202.61 | 171.99 | **15.1%** | 314.93 | 250.94 | **20.3%** | 729.42 | 458.81 | **37.1%** |
| | | 4.5 | 142.93 | 125.35 | **12.3%** | 186.25 | 158.68 | **14.8%** | 396.36 | 262.39 | **33.8%** |
| | | 5.5 | 120.09 | 108.68 | **9.5%** | 143.46 | 127.40 | **11.2%** | 252.51 | 188.12 | **25.5%** |
| | | 6.5 | 109.45 | 101.35 | **7.4%** | 124.40 | 114.03 | **8.3%** | 187.93 | 154.29 | **17.9%** |
| | | 8 | 102.53 | 96.28 | **6.1%** | 113.74 | 106.46 | **6.4%** | 153.56 | 134.52 | **12.4%** |
| | | 10.5 | 98.27 | 93.16 | **5.2%** | 104.36 | 99.04 | **5.1%** | 116.79 | 107.21 | **8.2%** |
| | | 13 | 96.57 | 92.42 | **4.3%** | 99.47 | 95.39 | **4.1%** | 108.12 | 102.61 | **5.1%** |

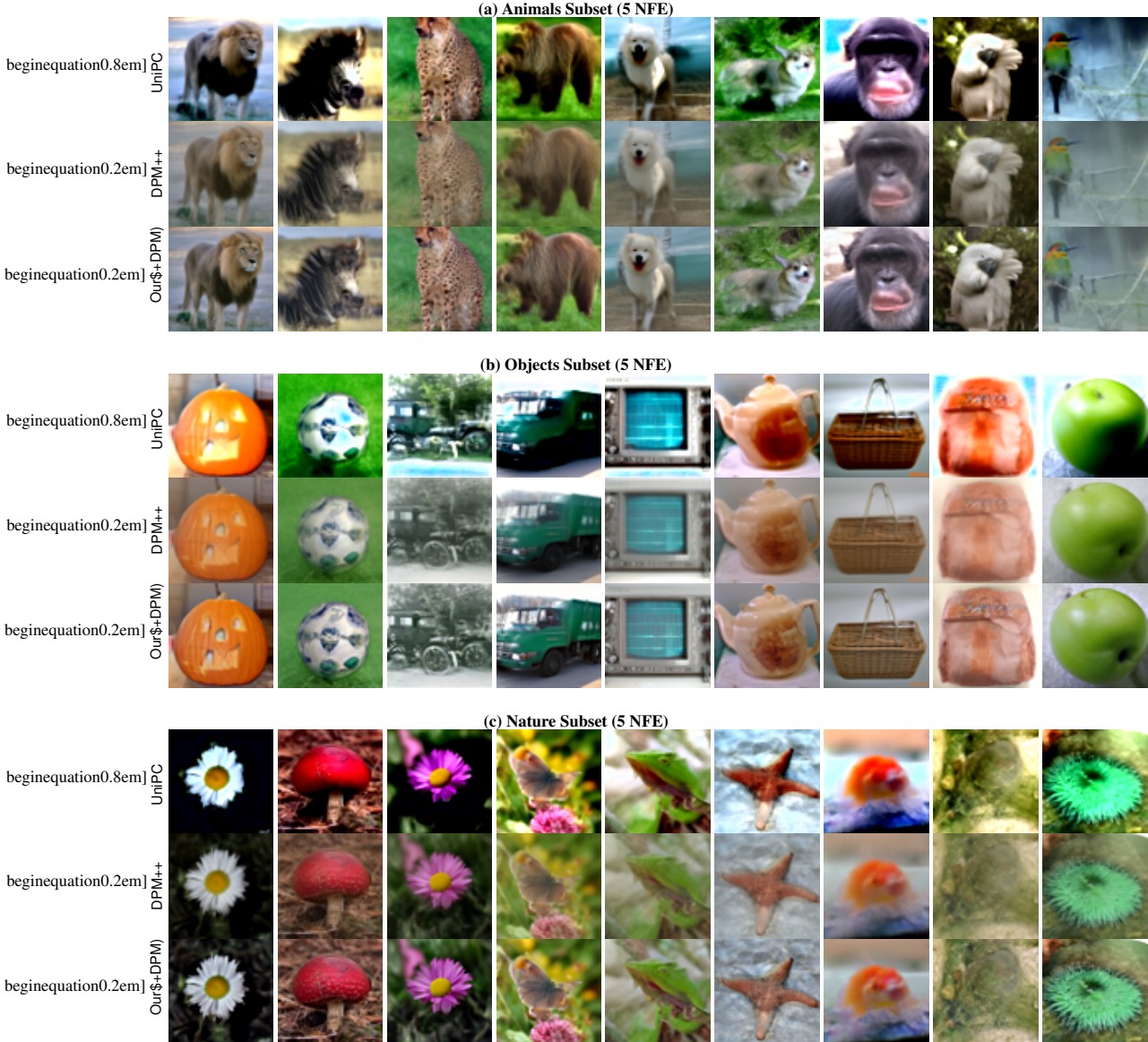

*Figure 10.* Mitigating the contractivity trap at extreme sparsity (5 NFE). While Efficient solvers like UniPC and DPM-Solver++ (DPM++) suffer from severe structural collapse and artifacts due to insufficient contractivity at large steps, SteinDiff (Our+DPM) successfully stabilizes the inference trajectory of efficient ODE solving. By explicitly correcting the geometric drift, our method preserves semantic fidelity across diverse categories (Animals, Objects, Nature).

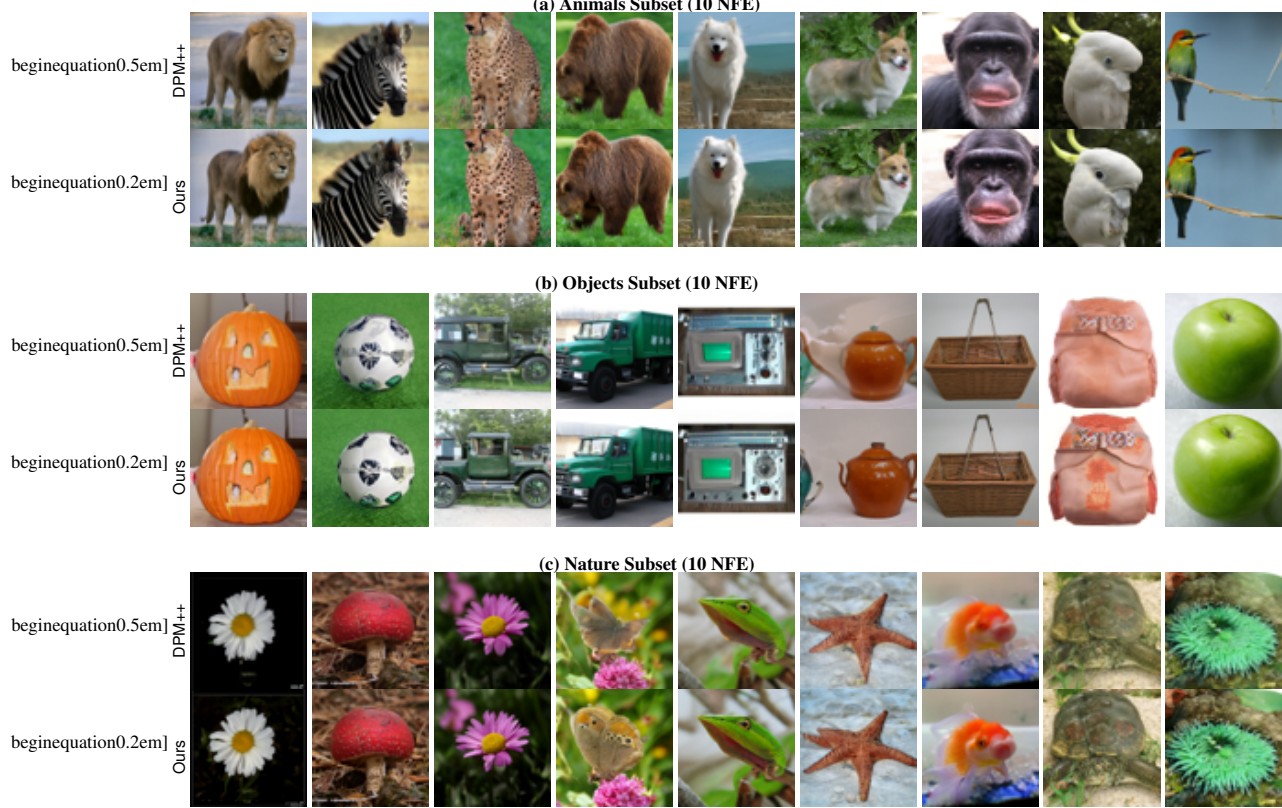

*Figure 11.* Enhanced fine-grained detail reconstruction at 10 NFE. Comparison between the baseline DPM-Solver++ (DPM++) and our SteinDiff-regularized version (Our). Even when the baseline achieves convergence, SteinDiff significantly refines high-frequency textures (e.g., animal fur, flower petals) and sharpens object boundaries. This demonstrates that our reference-free Stein stabilization improves perceptual quality by adhering closer to the data manifold.

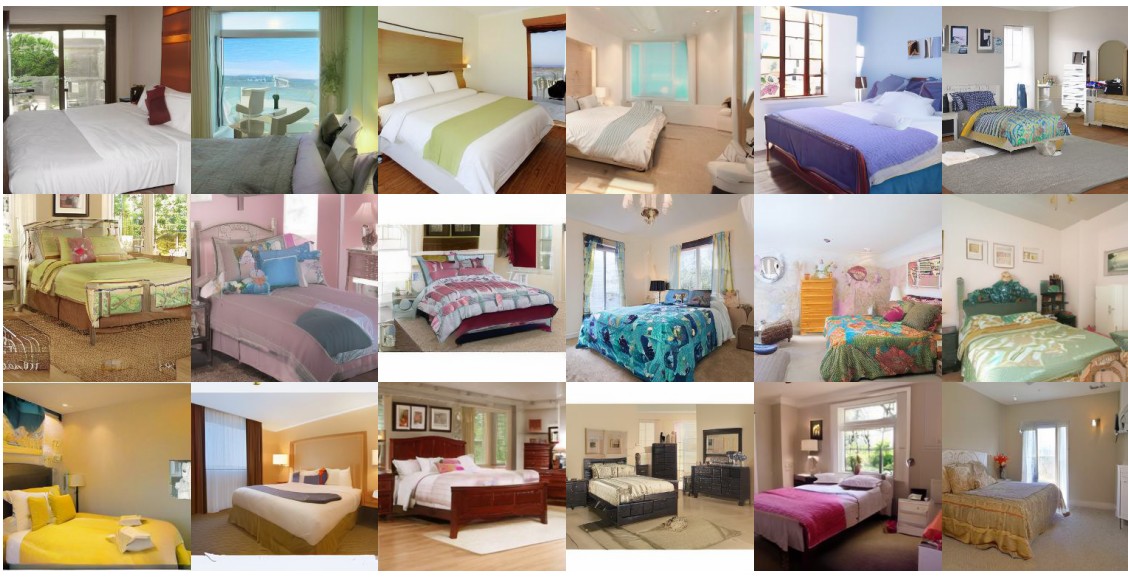

*Figure 12.* Samples generated by using SteinDiff for efficient DPM-Solver++ solving on LSUN Bedroom at 20 NFE. Achieving a SOTA FID of 2.77, these results empirically validate that our Hutchinson-based trace estimation remains robust and effective in high-dimensional latent spaces, effectively countering concerns regarding scalability and approximation errors.

