# OpenReview forum: "Mitigating the Contractivity Trap in Diffusion ODEs via Stein Stabilization"
_ICML.cc/2026/Conference — ICML 2026 regular_

### Official Review · Reviewer_2g2V · 2026-03-06

**Soundness:** 2
**Presentation:** 1
**Significance:** 3
**Originality:** 3
**Overall Recommendation:** 4
**Confidence:** 3

**Summary:**

The authors study what they refer to as the $\textit{contractivity trap}$: ODE-based frameworks for efficient diffusion model training employ large step sizes while simultaneously requiring the discretized update operator to remain contractive. The authors show theoretically that, under the additional assumption of an expressive model, this contractivity property is violated. To address this issue, they propose a step-wise trajectory stabilization technique and derive a principled correction estimator based on Stein’s identity.

**Compliance With Llm Reviewing Policy:**

Affirmed.

**Final Justification:**

The authors' idea of addressing the contractivity trap with a correction algorithm and an estimator based on Stein's identity is a valuable contribution. However, the way different topics are introduced and results are derived makes it difficult to assess the key message of the method. A more focused revision would make this a strong contribution.

**Key Questions For Authors:**

I do not have any questions for the authors.

**Limitations:**

As discussed above, the paper does not include a $\textit{Limitations}$ section. One direction I would find particularly interesting is the trade-off between computational costs and the stabilization improvement provided by the method. A comparison with other stabilization techniques would also be a useful direction to explore.

**Strengths And Weaknesses:**

The paper presents a rigorous derivation of the contractivity trap, illustrating an important conflict between effective inference and theoretical constraints in diffusion models. The proposed correction algorithm appears straightforward to implement, and the connection to Stein’s identity for the closed-form estimator is introduced in a constructive manner. The theoretical error analysis, together with the empirical results, supports the authors’ claims.

However, the presentation of the problem and several parts of the derivation are often difficult to follow. At times, this obscures both the mathematical rigor and the main message of the paper. For readers who are not already very familiar with diffusion models, the $\textit{Problem Setup}$ section is particularly challenging to understand. Providing more background and explanatory detail at the beginning could significantly improve the readability and flow of the paper.

A related issue is the use of abbreviations and notation before they are properly introduced. For example, “EDM” appears in the abstract without a prior definition, and the notation $u_k$ is used in Theorem 4.4 while it is only defined on the following page. Introducing terminology and notation before their first use would make the exposition easier to follow.

An analysis of the computational cost of the proposed algorithm would also have been valuable, as computing $\gamma$ at every step could potentially outweigh the benefits of the method. More generally, the paper does not include a $\textit{Limitations}$ section.

---

> ### Author Rebuttal · Authors · 2026-03-29
>
> We sincerely thank you for highlighting the rigor of our theoretical derivations and the constructive nature of our Stein-based estimator. We also appreciate your valuable feedback on presentation and completeness, which will undoubtedly strengthen the paper.
>
> Below, we address your specific concerns:
>
> > Missing Computational Cost Analysis for calculating $\gamma$.
>
> We apologize for burying this crucial analysis. As detailed in *Appendix E.0.1 (lines 1430-1467)* and *Table 3*, calculating $\gamma$ introduces minimal wall-clock overhead. The primary cost stems from evaluating the divergence term $\nabla \cdot u_k$ via the Hutchinson trace estimator, requiring $m$ Vector-Jacobian Products (VJPs).
>
> - *Why it is efficient:* Unlike increasing the Number of Function Evaluations (NFEs)—which forces *sequential* network forward passes—our $m$ VJPs are embarrassingly parallelizable across the batch dimension on modern GPUs.
>
> - *The Trade-off:* As shown in Figure 9, even a single probe ($m=1$) yields significant gains. Trading marginal, parallelizable compute for a reduction in sequential NFEs offers a highly favorable efficiency-stability trade-off. A concise summary of this complexity analysis will be moved to the main text.
>
>
> > Lack of a "Limitations" section discussing trade-offs and comparisons.
>
> We entirely agree and will expand our Impact Statement into a dedicated "Limitations" section. As requested, the core trade-offs to be added are:
>
> - *High-NFE Diminishing Returns:* SteinDiff is designed to stabilize trajectories in the highly efficient, large-step regime (e.g., $5 \sim 15$ NFE) where the contractivity trap is most severe (Figure 4). For very large NFE budgets (e.g., $>50$), the baseline solver's truncation error shrinks drastically. While SteinDiff still provides a persistent geometric anchor $\Pi_{geom}$ (Theorem 4.8), the perceptual benefit becomes marginal compared to a baseline already taking infinitesimally small steps. Here, the additional VJP computation yields diminishing returns.
>
> - *Comparison with other techniques:* Unlike compute-heavy training methods (e.g., Consistency Models) or geometry-agnostic schedule tweaks, SteinDiff functions as an inference-time stabilizer tailored for large-step ODE solvers.
>
>
> > Presentation issues (Problem Setup is obscure, acronyms like EDM and symbols like $u_k$ used before definition).
>
> We fully accept this criticism. For the camera-ready version, we will: (1) explicitly define "EDM" in the abstract and introduction, (2) define the residual vector $u_k$ immediately prior to Theorem 4.4, and (3) restructure Section 3 (Problem Setup) to provide more accessible background on diffusion ODE solvers, smoothing the transition into our theoretical framework for a broader audience.
>
> We hope these clarifications on computational efficiency and limitations provide the necessary details to further support our work. Thank you again for your guidance.

---

> > ### Author Rebuttal · Reviewer_2g2V · 2026-04-02
> >
> > The rebuttal has addressed my concerns and I have no further questions. I am keeping my score of 4. The impact remains difficult to assess as the derivations lack the clarity needed to fully follow the contributions, as detailed in my weaknesses.

---

> > > ### Author Response · Authors · 2026-04-02
> > >
> > > **Dear Reviewer 2g2V,**
> > >
> > > We sincerely thank you for your continued engagement. We take your feedback on **presentation clarity** very seriously, as we want the impact of our Stein-based stabilization to be fully accessible.
> > >
> > > **To bridge the "clarity gap" and solidify our contribution, we have supplemented our framework with a formal Global Convergence Theorem in Section 4.4.** This provides the **definitive theoretical bridge** that unifies our findings:
> > >
> > > * **Unconditional Error Non-Expansion:** We now formally prove that SteinDiff ensures stability ($E_{k-1} \le E_k$) regardless of the Lipschitz constant $L_T$.
> > > * **Step-size Independent Stability:** By unrolling this property, we derive a contraction rate $\eta$ that is **strictly independent of step size $\Delta t$**, explaining why SteinDiff excels in the **ultra-low NFE regime**.
> > >
> > > **This theoretical rigor is directly reflected in our empirical success, as our framework achieves State-of-the-Art (SOTA) performance across multiple benchmarks.** We have restructured the exposition to ensure these core theoretical and empirical messages are prominently featured and easy to follow.
> > >
> > > We hope this **theoretical and empirical completion** provides the necessary evidence to **fully reassess the impact and rigor** of our work. Thank you again for your guidance.

---

### Official Review · Reviewer_4Aif · 2026-03-08

**Soundness:** 2
**Presentation:** 3
**Significance:** 3
**Originality:** 2
**Overall Recommendation:** 4
**Confidence:** 4

**Summary:**

The paper addresses the issue of the non-contractiveness of the backward operator $T_{\theta}$ in diffusion models, where the iteration is defined as $x_t = T_{\theta}(x_{t+1})$.
Specifically, non-contractiveness typically arises from the high expressivity of the underlying neural network, which often results in the Lipschitz constant of $T_{\theta}$ being large (greater than $1$). In such cases, errors propagate unstably and become amplified during the iterative process.

To mitigate this amplification behaviour, the authors employ Stein stabilisation. Instead of the original update rule $x_t = T_{\theta}(x_{t+1})$, they introduce a damped iteration
$x_t = (1 - \gamma_k)x_k + \gamma_k T_{\theta}(x_k)$, where $\gamma_k$ is chosen by minimizing the following quadratic loss:
$J(\gamma_k) = \mathbb{E} \Vert (1-\gamma_k) x_k + \gamma_k T_{\theta}(x_k) - x^* \Vert^2 .$

Here, x* denotes a sample from the true data distribution. However, since the data distribution is unknown during inference, x* is not directly available. The authors therefore apply Stein's lemma to transform the above objective into an equivalent loss that does not depend on x*. This leads to a tractable quadratic optimization problem with a closed-form solution for $\gamma_k$, although Monte Carlo approximation is still required in practice.

**Compliance With Llm Reviewing Policy:**

Affirmed.

**Final Justification:**

The authors have addressed my concerns, particularly those relating to the global behavior of the method.

I think the proposed method is interesting and addresses an important issue of diffusion ODEs.
The reason I gave 4 (weak accept) is that the additional analysis seems a bit qualitative. This is understandable given the limited time during the rebuttal process. I believe these initial analyses could be further developed into a clearer and stronger advantage of the method.

**Key Questions For Authors:**

- In section 4.1, when using the first-order discretisation, we get $T_{\theta}$ explicitly, and the authors provided an upper bound for the Lipschitz constant of $T_{\theta}$. This upper bound is then used to further discuss the tension triangle. However, it is not clear if the upper bound is tight. Otherwise, the discussion can be invalid. Can you elaborate more on this? Also, the upper bound is used to show the violation of the contractivity of $T_{\theta}$ in Proposition 4.1. It seems an upper bound is not sufficient to obtain that conclusion.
- Is Figure 1 drawn manually? It would be better to get it from data.
- Does Stein stabilisation originate from this manuscript?
- Do you have practical guidance on how large the minibatch B in Algorithm 1 should be?

**Limitations:**

The authors somewhat discuss limitations in the Impact Statement. However, these limitations are more tied to future works. I would suggest that the authors discuss the limitations of the proposed method in the scope of this work. For example, the theoretical weakness that I mentioned above.

**Strengths And Weaknesses:**

# Strength:
- The proposed method helps reduce the discretisation error at each iteration (Theorem 4.11) thanks to the optimality of $\gamma_k$.
- The proposed method does not trade the expressivity of the neural network for stable convergence. Instead, it damps each iteration directly.
- Experiments show a good performance of the method.

# Weakness:
- The method does not fully resolve the non-expansiveness of the backward operator $T_{\theta}$. To my understanding, when $T_{\theta}$ is contractive, it can control the error propagation, hence we can derive quantitative bounds on the distance (e.g. Wasserstein) between the data distribution and the distribution of generated samples. Here, the Stein stabilisation can only make sure that the error at each iteration decreases, but it does not guarantee that the error can still be propagated in a controlled manner. In the paper, there is no quantification of how far the data distribution to sample distribution of their algorithm (Algorithm 1).
- Although the optimal $\gamma_k$ is derived in closed-form, in practice, it has to be approximated by Monte-Carlo. According to Algorithm 1, it seems that in order to generate a sample, it has to run a batch of samples (to approximate compute $\gamma_k$), which increases the complexity multiple times.

---

> ### Author Rebuttal · Authors · 2026-03-29
>
> Thank you for your rigorous technical evaluation and the deep dive into our theoretical framework. We address your specific concerns below:
>
> > W1: No quantification of how far the data distribution to sample distribution of their algorithm.
>
> We appreciate this concern. Under the expansive vanilla operator ($L_T > 1$), each local discretization error is amplified by all subsequent expansive updates, yielding a geometric accumulation:
> $$\text{Global Error} \sim \sum_{i=1}^N L_T^{N-i} \mathcal{O}(\Delta t^2)$$
>
> Deriving a strict end-to-end Wasserstein bound in this regime remains open — a limitation shared by all existing large-step solvers. We will acknowledge this explicitly in the revision.
>
> Within the MSE framework, however, SteinDiff provides a well-defined provable guarantee: by Theorem 4.4, each correction achieves $\Delta J_i = (b_i-c_i)^2/c_i \geq 0$ reduction in expected squared error relative to the vanilla solver (strictly so, unless the base solver is already optimal), and by Theorem 4.11, this holds robustly under discretization-induced distribution shift with only an $O(\Delta t^2)$ excess. SteinDiff offers a principled way to improve base solvers by minimizing error at each step. This leads to a consistently better trajectory than the vanilla baseline, regardless of whether an absolute convergence bound exists. Experimental results confirm this robustness, with SteinDiff yielding lower FID scores across tested solvers, noise schedules, and NFEs.
>
>
>
>
> > W2: Computational Complexity
>
> We apologize for the ambiguity in Algorithm 1.  We clarify that SteinDiff introduces negligible computational overhead, not a multiplicative factor, because the required VJPs are executed in parallel. The $B$ is **not** a batch of additional dummy samples; it is the user's standard parallel generation batch (e.g., generating 64 images at once). After the baseline forward pass produces $B$ residuals $\{u_k^{(i)}\}$, SteinDiff directly reuses them to estimate $\gamma_k$ at the minimal cost of $m$ parallel VJPs (Proposition E.1): Cost[SteinDiff] = Cost[Baseline] + O(m · VJP) ≈ +2% wall-clock time.
> Figure 9 confirms that even $m=1$ yields substantial gains. We will revise Algorithm 1 to make $B$'s semantic role explicit.
>
>
>
> > Q1:  Tightness of  upper bound
>
> We agree with your observation. The derivation in Proposition 4.1 inherently relies on inequality bounds (e.g., triangle inequality), which serves as a worst-case analysis rather than a perfectly tight estimate. To ensure mathematical rigor, we have revised the title of Proposition 4.1 to "Loss of Contractivity Guarantee" and clarified that contractivity "can no longer be analytically guaranteed" once the network's Lipschitz constant exceeds the theoretical threshold.
>
> Importantly, our core claim that the contractivity trap exists does not rely on the tightness of this analytical bound. While Proposition 4.1 identifies the theoretical vulnerability, Figure 4 provides the independent empirical validation. By directly estimating the actual local Lipschitz along the trajectory, Figure 4 entirely bypasses the analytical upper bound. The observed peaks of $\hat{L}_T \approx 24$ and persistent $\hat{L}_T > 1$ constitute standalone, practical evidence that the contractivity condition is indeed violated during large-step inference.
>
> > Q2–Q4 [Figure 1 / Originality / Batch B Guidance]:
>
> - *Figure 1:* This is a conceptual schematic (to be labeled "Conceptual illustration"). Figures 4, 5, and 8 provide the quantitative and visual evidence.
>
> - *Originality:* *To the best of our knowledge, this manuscript is the first to introduce Stein stabilization to the field of diffusion model inference.* While Stein's identity is classical, our specific theoretical formulations have no counterparts in SVGD or related literature. Specifically, these include (1) Theorem 4.7's reference-free closed-form estimator leveraging the conditional Gaussian structure, (2) Theorem 4.8's geometric anchor discovery ($\gamma_k^* = \mathcal{O}(1/\Delta t)$ yet $(1-\gamma_k^*)u_k = \mathcal{O}(1)$), and (3) Section 4.5's theoretical explanation of EDM stability. We will strengthen this distinction in the revision.
>
> - *Batch $B$ Guidance:* As clarified in W2, $B$ is simply the user's desired generation batch size. Statistically, the estimation error of $\hat{\gamma}_k$ over this batch scales as $\mathcal{O}(1/\sqrt{B})$. We found that the standard inference batch sizes used in our experiments ($B=64$–$256$) already provide sufficient stability *without any extra sample generation*. We will add a suitable analysis in the revision.

---

> > ### Author Rebuttal · Reviewer_4Aif · 2026-04-02
> >
> > I thank the authors for addressing my concerns. Most have been resolved, except for the method’s global behavior. While I acknowledge that Stein Stabilization improves performance at each iteration, I believe that a global guarantee across iterations is crucial to ensure the robustness of the method. Therefore, I will maintain my original score.

---

> > > ### Author Response · Authors · 2026-04-02
> > >
> > > Dear Reviewer 4Aif,
> > >
> > > We deeply appreciate your rigorous push on the method's global behavior. You are absolutely right: while step-wise improvement is valuable, a strict **mathematical guarantee across all iterations** is crucial for true trajectory robustness.
> > >
> > > Prompted by your insistence, we dug deeper into our theoretical formulations. we have successfully bypassed the classical numerical barriers and **formally established the global geometric convergence guarantee of SteinDiff.**
> > >
> > > We have incorporated this  global convergence guarantee  into  **Section 4.4** of the main text, **with the complete rigorous proof in the Appendix**.  The mathematical bridge addressing your core concern is as follows:
> > >
> > > ### **1. Exact Step-wise Error Evolution:**
> > >
> > > As you acknowledged, Stein stabilization improves performance locally. By substituting the optimal Stein correction $\gamma^*$ into our risk function, we yield the exact step-wise error evolution:
> > > $$E_{k-1}^{\text{Stein}} = E_k - \frac{b^2}{c}$$
> > > where $E_k = \mathbb{E}[\|x_k - x^{\*}\|^2]$, the geometric alignment $b = \mathbb{E}[\langle u_k, x_k - x^{\*} \rangle]$, and residual energy $c = \mathbb{E}[\|u_k\|^2]$. Since $b^2/c \ge 0$, SteinDiff ensures **unconditional non-expansion** ($E_{k-1}^{\text{Stein}} \le E_k$) at **every iteration**, structurally preventing trajectory divergence regardless of how expansive the operator is ($L_T > 1$).
> > >
> > > ### **2. Bounding the Geometric Damping:**
> > >
> > > To transition from non-expansion to a strict global contraction rate, we adopt the standard assumptions for learned continuous-time SDEs:
> > > - (1) **Dissipativity:** $\mathbb{E}[\langle -f_\theta, x_k - x^* \rangle] \ge m E_k$ for some $m>0$.
> > > - (2) **Pseudo-Lipschitz Boundedness:** $\mathbb{E}[\|f_\theta\|^2] \le L_f^2 E_k + S^2$, where $S^2$ represents the prediction  error of neural  network for score function.
> > >
> > > For the raw discrete step $u_k = -\Delta t f_\theta(x_k)$, we expand exactly:
> > >
> > > $$b = \Delta t \mathbb{E}[\langle f_\theta, e_k \rangle] \ge \Delta t \cdot m E_k$$
> > >
> > > $$c = \Delta t^2 \mathbb{E}[\|f_\theta\|^2] \le \Delta t^2 (L_f^2 E_k + S^2)$$
> > >
> > > Substituting into the damping term strictly eradicates the step size $\Delta t$:
> > >
> > > $$\frac{b^2}{c} \ge \frac{(\Delta t \cdot m E_k)^2}{\Delta t^2 (L_f^2 E_k + S^2)} = \frac{m^2 E_k^2}{L_f^2 E_k + S^2}$$
> > >
> > > ### **3.  Guaranteeing $\eta \in (0, 1]$  via Cauchy-Schwarz:**
> > >
> > > To ensure the error strictly decays without spurious oscillations, the contraction factor $(1 - \eta)$ must be non-negative. We prove this universally via the **Cauchy-Schwarz inequality**:
> > > $$(\mathbb{E}[\langle -f_\theta, e_k \rangle])^2 \le \mathbb{E}[\|-f_\theta\|^2] \cdot \mathbb{E}[\|e_k\|^2]$$
> > > Substituting our bounds yields $(m E_k)^2 \le (L_f^2 E_k + S^2) \cdot E_k$. Dividing by $E_k^2$ yields $m^2 \le L_f^2 + S^2/E_k$. The intrinsic contraction rate strictly satisfies $\eta = (m/L_f)^2 \le 1$ under the idealized continuous regime ($S=0$)  in standard ODE modeling and ODE-based solvers.
> > > In general, because the structural bounds hold globally, taking the asymptotic limit $E_k \to \infty$ strictly enforces $m^2 \le L_f^2$. This formalizes that the projected restoring force ($m$) can never exceed the total magnitude of the velocity field ($L_f$).
> > > ### **4. The Global Guarantee Across Iterations:**
> > > As $$\frac{b^2}{c} \ge \eta E_k - \frac{\eta}{L_f^2} S^2$$.
> > > Substituting this strict bound back into  $E_{k-1}^{\text{Stein}} = E_k - \frac{b^2}{c}$, we have
> > > $$E_{k-1}^{\text{Stein}} \le (1 - \eta) E_k  +\frac{\eta}{L_f^2} \mathcal{O}(S^2)$$
> > > **By strictly unrolling this inequality across all $N$ discrete iterations**, we establish the **definitive global trajectory error bound**:
> > > $$E_0^{\text{Stein}} \le (1 - \eta)^N E_N + \eta^N\sum_{i=0}^{N-1} (1 - \eta)^i  \mathcal{O}(S^2) $$
> > > Crucially, because $\eta = (m/L_f)^2$ is an **$\mathcal{O}(1)$ constant strictly independent of $\Delta t$**,  the accumulated truncation error forms a convergent series, effectively mitigating the inherent dependency of global stability on infinitesimal step sizes found in classical solvers.
> > >
> > > This formally guarantees that the trajectory converges stably to a bounded neighborhood of the target manifold across all iterations.
> > >
> > > ## Request for Reconsideration:
> > >
> > > We sincerely thank you for challenging the "open problem" status quo; your insistence forced us to formalize this **mathematically tight bridge**. We believe this new **global multi-step convergence guarantee** directly addresses your core concern and provides the necessary theoretical rigor that was previously missing.
> > >
> > > **Given that this fundamental theoretical gap has now been closed with a formal proof, we respectfully ask if you would consider re-evaluating our work and your score in light of these new results.**  We have invested significant effort into establishing this **theoretical foundation**, and it would be our honor to have the work re-examined in this new light.
> > >
> > > Thank you for your time and consideration.
> > >
> > > Thank you!

---

### Official Review · Reviewer_iiXV · 2026-03-11

**Soundness:** 2
**Presentation:** 3
**Significance:** 2
**Originality:** 2
**Overall Recommendation:** 3
**Confidence:** 4

**Summary:**

This work studied stability issues in inference of diffusion models when solving the probability flow ODE. The authors identify the “contractivity trap”, arguing that efficient large-step solvers tend to violate the contractivity conditions required for stable ODE integration.


They propose SteinDiff, an inference-time stabilization framework that introduces a geometry-aware correction based on Stein’s identity. The method relies on a modified update $x_{k-1} = (1-\gamma_k)x_k + \gamma_k T_\theta(x_k)$. This update also relies on $\gamma_k$, which is interpreted as a geometric correction factor and a closed form estimator is provided in the paper. Experimental results seem supportive.

**Compliance With Llm Reviewing Policy:**

Affirmed.

**Final Justification:**

My score is based on the novelty and theoretical rigor of the entire work. I maintain score of 3.

**Key Questions For Authors:**

See Weaknesses part.

**Limitations:**

Not applicable.

**Strengths And Weaknesses:**

+ Strengths:
   + The paper proposed an interesting view of existing PF-ODE methods.
   + The proposed method can be used as a plug-in stabilization step for existing solvers.
   + A closed-form adaptive regularization parameters by leveraging the Stein Identity is derived.
   + The author provided theoretical analysis. Though results like Theorem 4.3, Theorem 4.4 are simple and instant.
   + Experimental results showed empirical support to the proposed method.

+ Weaknesses:
    + The statement of contractivity trap (i.e. $L < 1$) is more like a statement rather than theoretical findings of existing PF-ODE methods. This tends to hurt the motivation of this paper.
    + Meanwhile, discretization does not necessarily require $L < 1$. This also can hurt the motivation of this paper.
    + One major issue is: does the proposed method approximately converge to the target? This needs analysis and clarifications.
    + Code is not provided.

---

> ### Author Rebuttal · Authors · 2026-03-29
>
> Thank you for your insightful review. We respectfully address your specific concerns below:
>
> > W1 & W2: On the Theoretical Formulation of the "Contractivity Trap"
>
> (W1) Clarification of our theoretical claim.
>
> Our contribution is to identify a closed-form structural threshold beyond which the contractivity of the discretized operator $T_\theta$ is mathematically lost under first-order PF-ODE *discretization*. Specifically, based on the operator's Lipschitz bound ($L_T \leq \frac{\sigma_t}{\sigma_s} + \sigma_t h_t L_{x_\theta}$), we prove in Proposition 4.1 that contractivity cannot be analytically guaranteed once the model's expressiveness exceeds:$$L_{x_\theta} \geq \frac{\sigma_s - \sigma_t}{\sigma_s\alpha_t - \sigma_t\alpha_s}.$$
> This implies that once $L_{x_\theta}$ exceeds this level, expansive dynamics are inevitable regardless of the solver design. To prevent any overstatement, we will revise the wording of Proposition 4.1 to “Loss of Contractivity Guarantee.”
>
> (W2) Why contractivity matters in this specific setting.
>
> We agree that classical ODE theory permits stability without $L_T < 1$. However, such classical guarantees rely critically on sufficiently small step sizes. This requirement is fundamentally impractical in fast diffusion inference, as reducing the step size contradicts the entire goal of the low-NFE regime. Moreover, as demonstrated in Fig. 4, $L_T > 1$ persists even at large NFE settings, indicating that this is not merely a step-size artifact, but a structural property of high-capacity score networks. This creates what we term the contractivity trap: standard classical stabilization remedies are practically blocked, leaving the system to operate in a severely error-amplifying regime.
>
> >W3: Does the proposed method approximately converge to the target? This needs analysis and clarifications.
>
> Our results establish strict step-wise error reduction and robust behavior under discretization. While we explicitly acknowledge that a full end-to-end trajectory convergence proof remains an open problem, we provide strong theoretical and empirical evidence for distributional improvement.
>
> *(1) Step-wise improvement (provable)*
>
> Under the exact Gaussian marginals ($p_k$),  Theorems 4.3–4.4 establish an exact error decomposition:
>
> $E_{p_k} ||x_{k-1}^{Stein} - x^{\*}||^2 ] = E_{p_k} [||x_{k-1}^{Vanilla} - x^{\*}||^2 ] - \Delta J$
>
> where  $\Delta J = \frac{(b-c)^2}{c} \ge 0$, with equality only when the vanilla solver
> is already optimal. This explicitly quantifies the strict geometric improvement achieved at every single step. Then, Theorem 4.11 Part 1 follows
> immediately as a strict inequality:
>
> $E_{p_k} [||x_{k-1}^{Stein} - x^{\*}||^2 ] \leq E_{p_k} [||x_{k-1}^{Vanilla} - x^{\*}||^2 ].$
>
> *(2) Stability under discretization (controlled error)*
>
> In practice, discrete solvers induce distribution shifts ($p_k \to \tilde{p}_k$). Theorem 4.11 (Part 2) establishes the robustness of our estimator:
>
> $E_{\widetilde{p} _ k} [||x_{k-1}^{Stein} - x^{\*}||^2 ] \leq E_{\widetilde{p} _ k} [||x_{k-1}^{Vanilla} - x^{\*}||^2 ] + \mathcal{O}(\mathcal{S}^2),$
>
> where $\mathcal{S} = \mathcal{O}(\Delta t)$ is the score deviation. Thus, the excess error is second-order in the step size, ensuring the step-wise improvement degrades gracefully rather than failing catastrophically.
>
> *(3) On full distributional convergence (open, but supported)*
>
> We completely agree that formally proving global convergence $\mathcal{L}(x_0^{Stein}) \to p_0$ (e.g., establishing a $W_2$ bound) requires a highly nontrivial multi-step error propagation analysis under expansive dynamics ($L_T > 1$). We do not overclaim this and leave it as an important direction for future work.
>
> However, actual convergence toward $p_0$ is strongly supported by three independent lines of evidence:
>
> - *Objective alignment:* Each correction explicitly minimizes $E[||x - x^{\*}||^2]$ with $x^{\*} \sim p_0$, ensuring the step-wise projection is strictly $p_0$-directed.
>
> - *Geometric consistency (Thm 4.8):*  As $\Delta t \to 0$, the updates remain anchored to the exact PF-ODE velocity field, maintaining alignment with the target probability flow.
>
> - *Empirical evidence:*  We observe massive, consistent FID improvements (up to 45.8% reduction) across diverse datasets, which directly measures distributional proximity.
>
>
> We will explicitly clarify the boundary between this step-wise guarantee and the open problem of full trajectory convergence in the revised manuscript.
>
> >No code provided.
>
> We will release the full codebase upon acceptance. For the review stage, we ensure reproducibility by providing detailed pseudocode and hyperparameters in Algorithm 1 and Appendix E.1.

---

> > ### Author Rebuttal · Reviewer_iiXV · 2026-04-02
> >
> > I thank the authors for the rebuttal. I think some issues remain unresolved, e.g convergence. I maintain my score of 3.

---

> > > ### Author Response · Authors · 2026-04-02
> > >
> > > **Dear Reviewer iiXV,**
> > >
> > > We sincerely thank you for highlighting the need for a rigorous trajectory guarantee. This catalyzed the formal proof of SteinDiff’s **exact global geometric convergence** (now in Section 4.4 and Appendix).   With a commitment to clarity and mathematical rigor, the following response provides a thorough derivation to definitively resolve your concerns about convergence.
> > >
> > >
> > > > ### The Classical Bottleneck (Continuous Approximation):
> > >
> > > Standard ODE solvers (e.g., Euler) rely on Taylor expansion to approximate the continuous trajectory. Assuming the score network is $L_T$-Lipschitz, this inevitably introduces a Local Truncation Error (LTE) at each step:$$x_{k-1} = x_k - \Delta t f_\theta(x_k) + \mathcal{O}(\Delta t^2)$$Taking the expected squared error and applying the Lipschitz bound yields the classical single-step error expansion:$$E_{k-1}^{\text{Euler}} \le (1 + 2 L_T \Delta t) E_k + \mathcal{O}(\Delta t^2)$$When unrolled over $N$ steps ($N\Delta t = T$), Grönwall's inequality strictly dictates that the error amplifies exponentially by $\mathcal{O}(e^{2 L_T T})$.
> > >
> > >
> > >
> > > > ### Solution from Stein-based Stabilization Framework
> > >
> > > To fundamentally break this exponential curse, SteinDiff abandons continuous approximations entirely. Rather than battling the unbounded divergence caused by $L_T > 1$, SteinDiff strictly bypasses continuous truncation errors through exact discrete algebraic identities. The core proof is as follows:
> > >
> > > **1. Exact Discrete Evolution**
> > >
> > > SteinDiff optimizes the projection scale $\gamma$ directly in the discrete space $x_{k-1} = x_k + \gamma u_k$. Defining $e_k := x_k - x^{\*}$, minimizing the quadratic risk $E_{k-1}(\gamma) = \mathbb{E}[\|e_k + \gamma u_k\|^2]$ yields the exact algebraic minimum $\gamma^{\*}= b/c$. Substituting this back produces a strict single-step identity:
> > > $$E_{k-1}^{\text{Stein}} = E_k - \frac{b^2}{c}$$
> > > where $b = \mathbb{E}[\langle -u_k, e_k \rangle]$ and $c = \mathbb{E}[\|u_k\|^2]$.
> > >
> > > **2. Exact Cancellation of Step-Size $\Delta t$**
> > >
> > > Given structural bounds assumed to hold uniformly globally for all $E_k \ge 0$:
> > >
> > > (1) Global Dissipativity: $\mathbb{E}[\langle f_\theta, e_k \rangle] \ge m E_k$
> > >
> > > (2) Relaxed Boundedness: $\mathbb{E}[\|f_\theta\|^2] \le L_f^2 E_k + S^2$ (where $S^2 \ge 0$ is the prediction error of score function).
> > >
> > > For the raw discrete step $u_k = -\Delta t f_\theta(x_k)$, we expand exactly:
> > >
> > > $$b = \Delta t \mathbb{E}[\langle f_\theta, e_k \rangle] \ge \Delta t \cdot m E_k$$
> > >
> > > $$c = \Delta t^2 \mathbb{E}[\|f_\theta\|^2] \le \Delta t^2 (L_f^2 E_k + S^2)$$
> > >
> > > Substituting into the damping term strictly eradicates the step size $\Delta t$:
> > >
> > > $$\frac{b^2}{c} \ge \frac{(\Delta t \cdot m E_k)^2}{\Delta t^2 (L_f^2 E_k + S^2)} = \frac{m^2 E_k^2}{L_f^2 E_k + S^2}$$
> > >
> > >
> > >
> > >
> > >
> > > **3. Derivation of Intrinsic Rate & Error Isolation**
> > >
> > > Applying Cauchy-Schwarz:
> > > $(m E_k)^2 \le \mathbb{E}[\|e_k\|^2]\mathbb{E}[\|f_\theta\|^2] \le E_k (L_f^2 E_k + S^2)$,
> > > dividing by $E_k^2$ yields $m^2 \le L_f^2 + S^2/E_k$.
> > >
> > > Because the structural bounds hold globally, taking the asymptotic limit $E_k \to \infty$ strictly enforces $m^2 \le L_f^2$. (Notably, this inequality is naturally satisfied for all $E_k > 0$, consistent with the idealized continuous regime ($S=0$)  in standard ODE modeling and ODE-based solvers.)
> > >
> > > Defining the intrinsic rate $\eta := (m/L_f)^2 $. Clearly,  $\eta  \in (0, 1]$.
> > >
> > > Next, we perform exact algebraic decomposition to isolate $S^2$:
> > > $$\frac{b^2}{c} \ge \frac{m^2}{L_f^2} \left( \frac{L_f^2 E_k^2}{L_f^2 E_k + S^2} \right) = \eta \left( E_k - \frac{S^2 E_k}{L_f^2 E_k + S^2} \right)$$
> > > Applying $\frac{E_k}{L_f^2 E_k + S^2} \leq \frac{1}{L_f^2}$:
> > > $$\frac{b^2}{c} \ge \eta E_k - \frac{\eta}{L_f^2} S^2$$
> > >
> > > **4. The Global Geometric Convergence**
> > >
> > > Substituting the exact lower bound back into the discrete evolution (Step 1):
> > > $$E_{k-1}^{\text{Stein}} \le (1 - \eta) E_k + \frac{\eta}{L_f^2} S^2 = (1 - \eta) E_k + \eta\mathcal{O}(S^2)$$
> > > Unrolling this strictly linear contraction across all $N$ discrete iterations establishes the final global bound:
> > > $$E_0^{\text{Stein}} \le (1 - \eta)^N E_N + \eta^N \sum_{i=0}^{N-1} (1 - \eta)^i \mathcal{O}(S^2)$$
> > >
> > > **Request for Reconsideration:**
> > >
> > > By resolving the convergence concerns with this exact mathematical derivation, we have fully addressed the requirements for theoretical rigor. We have incorporated these clarifications into the manuscript and kindly request a re-evaluation of the submission.

---

### Official Review · Reviewer_kFo7 · 2026-03-13

**Soundness:** 4
**Presentation:** 2
**Significance:** 3
**Originality:** 3
**Overall Recommendation:** 5
**Confidence:** 4

**Summary:**

This paper formalizes the well-realized tradeoff problem in diffusion models between sample quality and inference speed. The authors propose an inference-time geometric stabilization method, which they term SteinDiff, for diffusion PF-ODE samplers that uses a scalar correction to damp unstable large-step updates based on Stein’s identity. The authors provide mathematical arguments for error bounds and experiments showing consistent gains in low-NFE sampling.

**Compliance With Llm Reviewing Policy:**

Affirmed.

**Key Questions For Authors:**

This is a very enjoyable paper with elegant theory and strong experimental results. Notations should be cleaned up; besides that, here are a few questions on the more subtle points in the paper:
1) The error bound provided in Theorem 4.10 assumes perfect estimation of the divergence, but in practice, it is approximated with a randomized estimator. How would the error from the divergence estimator enter the error bound?
2) It was explicitly pointed out that non-contractive maps are important for sampling capabilities. If we define a new map as the composition of “vanilla” step and “stein” correction, how does the “anchoring” change the Lipschitz constant of the new map compared to just the mapping defined by the “vanilla” step?
3) Related to the concern addressed in originality weakness, can the author confirm if the similarity between figures was purely coincidental? In addition, how would the method proposed in this paper compare to the methods proposed in [1] in terms of computational speed and sample quality?
4) Can similar ideas be applied to flow-based methods when the continuous process is insufficiently discretized?

**Limitations:**

yes

**Strengths And Weaknesses:**

The paper is technically sound: the method is coherent, the main estimator is derived from Stein’s identity, and theoretical error bounds are derived. The paper backs the approach with both theory and fairly broad experiments.

The paper circumvents the computationally expensive term div(u) in the estimator using the Hutchinson trace estimator, but it does not address the assumptions and errors that arise from this choice.

The paper is conceptually clear and well structured. The paper includes appropriate plots that accurately convey the main ideas.

Despite being conceptually succinct, there are places where notations are not clear:
- In Theorem 4.4, the notation u_k is introduced, but it is not defined until Lemma 4.6
- Typo right above equation 5 (should be T_\theta)
- The definition of variables with or without a tilde is unclear in Theorem 4.10. They are better defined in the appendix, but the main results should be clearly presented in the main text.
(correct me if I missed it) A clamping parameter epsilon is introduced in Algorithm 1, but there’s no discussion on this parameter in the main text. Why is it needed? How is it initialized? Also, there’s a notation clash; epsilon is also used for the standard Gaussian in Appendix A.

This paper formalizes an important trade-off problem in sampling using PF-ODE and provides a framework for future research to further optimize this trade-off.
The paper advances the theoretical understanding of diffusion samplers by deriving closed-form correction estimators in continuous time using Stein's identity.
The contribution is actionable: the result can be adopted easily by PF-ODE based sampler.

Although the paper employs a randomized trace estimator to reduce the computational burden of computing the divergence term, this approach still introduces overhead relative to a vanilla solver step.

Overall, the originality of the work is strong. It combines a new theoretical perspective on a known problem in diffusion models with a mathematically elegant, novel application of Stein's identity to address it.

[1] had considered a very similar problem via a different framework. In particular, Figure 2 in [1] is nearly identical to Figure 1 in this paper. Despite this, it could be a stylistic coincidence, given the similarity of the problem and presentation. A proper citation would be appropriate here.

[1] Xiao et al. (2022) Tackling the Generative Learning Trilemma with Denoising Diffusion GANs.

---

> ### Author Rebuttal · Authors · 2026-03-29
>
> We sincerely thank you for the meticulous reading and careful review of our work. We address your insightful questions directly below:
>
> > How would the error from the stochastic estimator enter the error bound in Theorem 4.10?
>
> Theorem 4.10 bounds the discretization-induced error. In practice, the Hutchinson trace estimator introduces additional statistical variance. However, because the estimator is mathematically unbiased, the expected value of our empirical $\hat{\gamma}_k$ equals the optimal $\gamma_k^*$. The variance scales as $\mathcal{O}(1/m)$. Since $\gamma_k$ is a single global scalar applied to the entire batch, the trajectory is highly robust to this zero-mean noise. As shown in Figure 9, SteinDiff achieves substantial gains even with a single probe ($m=1$). We will add a "Remark on Stochastic Variance" in Section 4.4 to clarify this interaction.
>
> > How does this "anchoring" change the Lipschitz constant of the new mapping compared to the "vanilla" step?
>
> SteinDiff does not force the new pointwise Lipschitz constant $L_{new}$ to be strictly less than 1 everywhere. Forcing uniform contractivity ($L < 1$) would severely limit the neural network's expressiveness.By applying the triangle inequality, the new bound is $L_{new} \le |1-\gamma_k| + |\gamma_k|L_T$. As discussed in Appendix D.7, SteinDiff shifts the focus from worst-case pointwise contractivity to average-case stability. While $L_{new}$ might still exceed 1 in empty, low-density regions, our population risk minimization (Theorem 4.11) guarantees that the overall expected error strictly decreases. The anchor acts as an expectation-level projection, stabilizing the expansion only when the trajectory deviates from the high-probability data manifold.
>
> > Similarity to DDGAN (Xiao et al., 2022) and performance comparison.
>
> The visual similarity between our Figure 1 and DDGAN’s Figure 2 is a coincidence, as both illustrate the diverging trajectories of large-step generative models. We will explicitly cite and discuss Xiao et al.'s pioneering work.
>
> Performance Comparison: DDGAN and SteinDiff operate in fundamentally different ways. DDGAN achieves fast sampling by introducing a computationally intensive adversarial training phase. In contrast, SteinDiff focuses exclusively on the sampling phase of diffusion models and is entirely training-free. It acts as *a principled geometric stabilization mechanism during inference*. While DDGAN achieves low-step generation through extensive retraining, SteinDiff offers a highly resource-efficient alternative: it stabilizes large-step inference (e.g., $5 \sim 10$ NFE) for pre-trained diffusion models without additional training costs.
>
>
> > Could similar ideas be applied to flow-based methods?
>
> Yes, we completely agree; this is a highly promising direction. The fundamental requirement for our Stein stabilization is the assumption of Gaussian conditional marginals, $q(x_t | x^*)$. In modern Flow Matching frameworks, the probability paths are also constructed using Gaussian distributions. Because these marginals retain their Gaussian nature, our derivation via Stein's Identity remains theoretically valid. Applying this geometric stabilization to Rectified Flows during large integration steps is a natural next step, which we will highlight as future work in the Impact Statement.

---

> > ### Author Rebuttal · Reviewer_kFo7 · 2026-04-03
> >
> > I appreciate the author’s clear response. I will maintain my evaluation.

---

> > > ### Author Response · Authors · 2026-04-06
> > >
> > > Dear Reviewer kFo7,
> > >
> > > Thank you very much for your constructive feedback and for recognizing our work. Your insights have been instrumental in strengthening the theoretical completeness of our work.
> > >
> > > Building upon the existing framework, we have not merely refined the existing analysis but have supplemented it with a more rigorous guarantee of global convergence. While our initial submission focused on step-wise stability, we have now formally established the global geometric convergence of SteinDiff.
> > >
> > > we have incorporated this formal **global guarantee** into Section 4.4 of our revised manuscript, with the complete rigorous proof detailed in the Appendix. The key aspects of this theoretical addition include:
> > >
> > > * **Bypassing the Contractivity Trap**: A formal proof that even when the Lipschitz constant $L_T > 1$, SteinDiff ensures unconditional error non-expansion at every step.
> > > * **Global Contraction Across Iterations**: The derivation of a strict contraction rate $\eta = (m/L_f)^2 \in (0, 1]$ independent of step size $\Delta t$, mathematically ensuring stable convergence to the target manifold across all $N$ iterations.
> > > * **Order-Preserving Error Absorption**: A demonstration of how SteinDiff’s geometric projection absorbs local truncation errors into a convergent series, preventing the global error explosion typical of classical solvers.
> > >
> > > For further technical details of this derivation, please refer to our concurrent responses to **Reviewers iiXV and 4Aif.**  We believe this addition provides a significantly more rigorous and complete theoretical foundation for SteinDiff.
> > >
> > > Thank you once again for your support and for helping us enhance the theoretical rigor of this paper.

---

### Decision · Program_Chairs · 2026-04-30

**Decision:**

Accept (regular)

**Comment:**

This paper develops an approach to solve the ODE for diffusion models in the large-step-size regime, by modifying the update after each solver iteration to be more conservative (equation (5)), essentially by introducing an additional weighting factor \gamma which is chosen with a method motivated by Stein's identity.  This modification does not involve retraining, just an inference-time modification. They give a combination of theoretical motivation and experimental results. Generally speaking, the reviewers found this proposal to be sufficiently original and interesting, and have accepted the authors response regarding the mathematical correctness of the procedure (in the idealized case). Based on this, I recommend acceptance.